# Learning Stochastic Majority Votes by Minimizing a PAC-Bayes Generalization Bound

**Valentina Zantedeschi**[123]  **Paul Viallard**[4]  **Emilie Morvant**[4]  **Rémi Emonet**[4]
**Amaury Habrard**[4]  **Pascal Germain**[5]  **Benjamin Guedj**[123]

[1] Inria, Lille - Nord Europe research centre, France  [2] The Inria London Programme, France and UK
[3] University College London, Department of Computer Science, Centre for Artificial Intelligence, UK
[4] Univ Lyon, UJM-Saint-Etienne, CNRS, Institut d Optique Graduate School,
Laboratoire Hubert Curien UMR 5516, F-42023, Saint-Etienne, France
[5] Département d'informatique et de génie logiciel, Université Laval, Québec, Canada

## Abstract

We investigate a stochastic counterpart of majority votes over finite ensembles of classifiers, and study its generalization properties. While our approach holds for arbitrary distributions, we instantiate it with Dirichlet distributions: this allows for a closed-form and differentiable expression for the expected risk, which then turns the generalization bound into a tractable training objective. The resulting stochastic majority vote learning algorithm achieves state-of-the-art accuracy and benefits from (non-vacuous) tight generalization bounds, in a series of numerical experiments when compared to competing algorithms which also minimize PAC-Bayes objectives – both with uninformed (data-independent) and informed (data-dependent) priors.

## 1 Introduction

By combining the outcomes of several predictors, ensemble methods [Dietterich, 2000] have been shown to provide models that are more accurate and more robust than each predictor taken singularly. The key to their success lies in harnessing the diversity of the set of predictors [Kuncheva, 2004]. Among ensemble methods, weighted Majority Votes (MV) classifiers assign a score to each base classifier (*a.k.a.* voter) and output the most common prediction, given by the weighted majority. When voters have known probabilities of making an error and make independent predictions, the optimal weighting is given by the so-called Naive Bayes rule [Berend and Kontorovich, 2015]. However, in most situations these assumptions are not satisfied, giving rise to the need for techniques that estimate the optimal combination of voter predictions from the data.

Among them, PAC-Bayesian based methods are well-grounded approaches for optimizing the voter weighting. Indeed, PAC-Bayes theory (introduced by Shawe-Taylor and Williamson, 1997, McAllester, 1999 – we refer to Guedj, 2019 for a recent survey and references therein) provides not only bounds on the true error of a MV through Probably Approximately Correct (PAC) generalization bounds (see *e.g.* Catoni [2007], Seeger [2002], Maurer [2004], Langford and Shawe-Taylor [2002], Germain et al. [2015]), but is also suited to derive theoretically grounded learning algorithms (see *e.g.* Germain et al. [2009], Roy et al. [2011], Parrado-Hernández et al. [2012], Alquier et al. [2016]). Contrary to the most classical PAC bounds [Valiant, 1984], as VC-dimension [Vapnik, 2000] or Rademacher-based bounds [Mohri et al., 2012], PAC-Bayesian guarantees do not stand for all hypotheses (*i.e.* are not expressed as a worst-case analysis) but stand in expectation over the hypothesis set. They involve a hypothesis space (formed by the base predictors), a prior distribution on it (*i.e.* an *a priori* weighting) and a posterior distribution (*i.e.* an *a posteriori* weighting) evaluated on a learning sample. The prior brings some prior knowledge on the combination of predictors, and the posterior

distribution is learned (adjusted) to lead to good generalization guarantees; the deviation between the prior and the posterior distributions plays a role in generalization guarantee and is usually captured by the Kullback-Leibler (KL) divergence. In their essence, PAC-Bayesian results do not bound directly the risk of the deterministic MV, but bound the expected risk of one (or several) base voters randomly drawn according to the weight distribution of the MV [Langford and Shawe-Taylor, 2002, Lacasse et al., 2006, 2010, Germain et al., 2015, Masegosa et al., 2020]. This randomization scheme leads to upper bounds on the true risk of the MV that are then used as a proxy to derive PAC-Bayesian generalization bounds. However, the obtained risk certificates are generally not tight, as they depend on irreducible constant factors, and when optimized they can lead to sub-optimal weightings. Indeed, by considering a random subset of base predictors, state-of-the-art methods do not fully leverage the diversity of the whole set of voters. This is especially a problem when the voters are weak, and learning to combine their predictions is critical for good performance.

**Our contributions.** In this paper, we propose a new randomization scheme. We consider the voter weighting associated to a MV as a realization of a distribution of voter weightings. More precisely, we analyze with the PAC-Bayesian framework the expected risk of a MV drawn from the posterior distribution of MVs. The main difference with the literature is that we propose a stochastic MV, while previous works aim at studying randomized evaluations of the true risk of the deterministic MV. Doing so, we are able to derive tight empirical PAC-Bayesian bounds for our model directly on its expected risk, in Section 4. We further propose, in Section 3 two approaches for optimizing the generalization bounds, hence learning the optimal posterior: the first optimizes an analytical and differentiable form of the empirical risk that can be derived when considering Dirichlet distributions; the second optimizes a Monte Carlo approximation of the expected risk and can be employed with any form of posterior. In our experiments, reported in Section 5, we first compare these two approaches, highlighting in which regimes one is preferable to the other. Finally, we assess our method's performance on real benchmarks *w.r.t.* the performance of PAC-Bayesian approaches also learning MV classifiers. These results indicate that our models enjoy generalization bounds that are consistently tight and non-vacuous both when studying ensembles of data-independent predictors and when studying ensembles of data-dependent ones.

**Societal impact.** Our work abides by the ethical guidelines enforced in contemporary research in machine learning. Given the theoretical nature of our contributions we do not foresee immediate potentially negative societal impact.

## 2 Notation and background

In this section, we formally define weighted Majority Vote (MV) classifiers and review the principal PAC-Bayesian approaches for learning them.

### 2.1 Weighted majority vote classifiers

Consider the data random variable $(X, Y)$, taking values in $\mathcal{X} \times \mathcal{Y}$ with $\mathcal{X} \subseteq \mathbb{R}^d$ a $d$-dimensional representation space and $\mathcal{Y}$ the set of labels. We denote $\mathcal{P}$ the (unknown) data distribution of $(X, Y)$. We define a set (dictionary) of base classifiers $D = \{h_j : \mathcal{X} \to \mathcal{Y}\}_{j=1}^M$. The weighted majority vote classifier is a convex combination of the base classifiers from $D$. Formally, a MV is parameterized by a weight vector $\theta \in [0, 1]^M$, such that $\sum_{j=1}^M \theta_j = 1$ hence lying in the $(M\text{-}1)$-simplex $\Delta^{M-1}$, as follows:

$$f_\theta(x) = \operatorname*{argmax}_{y \in \mathcal{Y}} \sum_{j=1}^M \theta_j \, \mathbb{1}(h_j(x) = y), \tag{1}$$

where $\mathbb{1}(\cdot)$ is the indicator function. Let $W_\theta(X, Y)$ be the random variable corresponding to the total weight assigned to base classifiers that predict an incorrect label on $(X, Y)$, that is

$$W_\theta(X, Y) = \sum_{j=1}^M \theta_j \mathbb{1}(h_j(X) \neq Y). \tag{2}$$

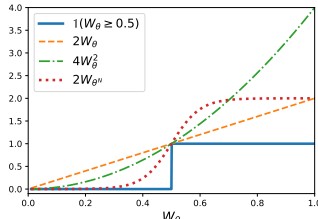

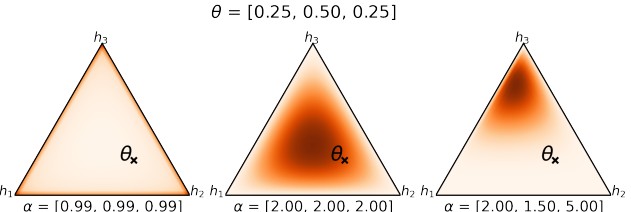

Figure 1: Oracle upper bounds for the true risk. The risks are the areas under the respective curves, for an arbitrary distribution of $W_\theta$ (typically different from the uniform).

Figure 2: Visualization of the density measure $\rho : \Delta^2 \to \mathbb{R}_+$ taking the form of a Dirichlet distribution, with concentration parameters $\alpha$. The darker the color, the higher $\rho(\theta)$. Each $\theta$ on the simplex corresponds to a majority vote classifier $f_\theta$ and has an associated probability depending on $\alpha$.

In binary classification with $|\mathcal{Y}|=2$, the MV errs whenever $W_\theta(X,Y) \geqslant 0.5$ [Lacasse et al., 2010, Masegosa et al., 2020]. Hence the true risk (*w.r.t.* 01-loss) of the MV classifier can be expressed as

$$R(f_\theta) \triangleq \mathbb{E}_\mathcal{P}\, \mathbb{1}(W_\theta(X,Y) \geqslant 0.5) = \mathbb{P}(W_\theta \geqslant 0.5). \qquad (3)$$

Similarly, the empirical risk of $f_\theta$ on a $n$-sample $S=\{(x_i,y_i)\sim\mathcal{P}\}_{i=1}^n$ is given by

$$\hat{R}(f_\theta) = \sum_{i=1}^n \mathbb{1}(W_\theta(x_i,y_i) \geqslant 0.5).$$

Note that the results we introduce in the following are stated for binary classification, but are valid also in the multi-class setting ($|\mathcal{Y}|>2$). Indeed, in this context Equation (3) becomes a surrogate of the risk: we have $R(f_\theta) \leqslant \mathbb{P}(W_\theta \geqslant 0.5)$ (see the notion of $\omega$-margin with $\omega = 0.5$ proposed by Laviolette et al. [2017]).

## 2.2 A PAC-Bayesian perspective on the majority vote

The current paper stands in the context of PAC-Bayesian MV learning. The PAC-Bayesian framework has been employed to study the generalization guarantees of randomized classifiers. It is known to provide tight risk certificates that can be used to derive self-bounding learning algorithms [*e.g.* Germain et al., 2009, Dziugaite and Roy, 2017, Pérez-Ortiz et al., 2020]. In the PAC-Bayes literature of the analysis of MV, $\theta$ is interpreted as the parameter of a categorical distribution $\mathcal{C}(\theta)$ over the set of base classifiers $D$ [*e.g.* Germain et al., 2015, Lorenzen et al., 2019, Masegosa et al., 2020]. In this sense, $f_\theta$ corresponds to the MV predictor $f_\theta(x)= \operatorname{argmax}_{y\in\mathcal{Y}} \mathbb{E}_{h\sim\mathcal{C}(\theta)}\mathbb{1}(h(x)=y)$ and $W_\theta$ corresponds to the expected ratio of wrong predictors $W_\theta(X,Y) = \mathbb{E}_{h\sim\mathcal{C}(\theta)}\mathbb{1}(h(X) \neq Y)$. The PAC-Bayesian analysis provides a sensible way to find such a categorical distribution, called *posterior* distribution, that leads to a model with low true risk $R(f_\theta)$. However, an important caveat is that the PAC-Bayesian generalization bounds cannot be derived directly on the true risk $R(f_\theta)$, without making assumptions on the distribution of $W_\theta$ and raising fidelity problems. Thus, a common approach is to consider oracle[1] upper bounds on the true risk in terms of quantities from which PAC-Bayesian bounds can be derived, that are typically related to the statistical moments of $W_\theta$. By doing so, oracle bounds act as a proxy for estimating the cumulative density function for $W_\theta \geqslant 0.5$ [Langford and Shawe-Taylor, 2002, Germain et al., 2006, Lacasse et al., 2010, Masegosa et al., 2020]. Generalization guarantees for $R(f_\theta)$ are hence derived, involving the empirical counterpart of the oracle bound, the KL-divergence between the *posterior* categorical distribution $\mathcal{C}(\theta)$ and a *prior* one. An overview of the existing oracle bounds described below is represented in Figure 1.

**First order bound.** The most classical "factor two" oracle bound [Langford and Shawe-Taylor, 2002] is derived considering the relation between MV's risk and the first moment of $W_\theta$, *a.k.a.* the expected risk of the randomized classifier:

$$R_1(\theta) \triangleq \mathbb{E}_{h\sim\mathcal{C}(\theta)}\mathbb{1}(h(X)\neq Y) = \mathbb{E}_\mathcal{P}\, W_\theta.$$

---

[1]Oracle bounds are expressed in terms of the unknown data distribution; their exact value cannot be computed.

By applying Markov's inequality, we have $R(f_\theta) \leqslant 2R_1(\theta)$. This "factor two" bound is close to zero only if the expectation of the risk of a single base classifier drawn according to $\mathcal{C}(\theta)$ is itself close to zero. Therefore, it does not take into account the correlation between base predictors, which is key to characterize how a MV classifier can achieve strong predictions even when its base classifiers are individually weak (as observed when performing *e.g. Boosting* [Schapire and Singer, 1999]). This explains why $R_1(\theta)$ can be a very loose estimate of $R(f_\theta)$ when the base classifiers are adequately decorrelated.

**Binomial bound.** A generalization of the first order approach was proposed in Shawe-Taylor and Hardoon [2009], Lacasse et al. [2010], where the true risk of the MV is estimated by drawing multiple ($N$) base hypotheses and computing the probability that at least $\frac{N}{2}$ make an error (which is given by the binomial with parameter $W_\theta$):

$$W_{\theta^N}(X, Y) \triangleq \sum_{k=\frac{N}{2}}^{N} \binom{N}{k} W_\theta^k (1 - W_\theta)^{(N-k)}.$$

The higher $N$, the better $W_{\theta^N}(X, Y)$ approximates the true risk, but the looser the bound, as the KL term worsens by a factor of $N$. Moreover, with this approach it is not possible to derive generalization bounds directly on the true risk, and the corresponding oracle bound still presents a factor two: $R(f_\theta) \leqslant 2 \, \mathbb{E}_\mathcal{P} W_{\theta^N}$.

**Second order bound.** A parallel line of works focuses on improving the bounds by accounting for voter correlations, *i.e.* considering the agreement and/or disagreement of two random voters. Masegosa et al. [2020] recently proposed a new upper bound for the true risk depending on the second moment of $W_\theta$, *a.k.a.* tandem loss or joint error, the risk that two predictors make a mistake on the same point:

$$R_2(\theta) \triangleq \mathbb{E}_\mathcal{P} \, \mathbb{E}_{h \sim \mathcal{C}(\theta), h' \sim \mathcal{C}(\theta)} \mathbb{1}(h(X) \neq Y \wedge h'(X) \neq Y) = \mathbb{E}_\mathcal{P} \, W_\theta^2.$$

By applying the second order Markov's inequality, we have $R(f_\theta) \leqslant 4R_2(\theta)$. Masegosa et al. [2020] show that in the worst case (*i.e.* when the base classifiers are identical) the second order bound could be twice worse than the first order bound ($R_2(\theta) \approx 2R_1(\theta)$), while in the best case (*i.e.* the $M$ base classifiers are perfectly decorrelated), the second order bound could be *an order of magnitude* tighter ($R_2(\theta) \approx \frac{1}{M} R_1(\theta)$).

**C-bound.** Originally proposed by Breiman [2001] in the context of Random Forest, a bound known as the C-Bound in the PAC-Bayesian literature [Lacasse et al., 2006, Roy et al., 2011, Germain et al., 2015, Laviolette et al., 2017, Viallard et al., 2021b] is derived by considering explicitly the joint error and disagreement between two base predictors. Using Chebyshev-Cantelli's inequality, the C-Bound can be written in terms of the first and second moments of $W_\theta$:

$$R(f_\theta) \leqslant \frac{R_2(\theta) - \mathbb{E}_{h \sim \mathcal{C}(\theta)} \left(\mathbb{E}_\mathcal{P} \mathbb{1}(h(X) \neq Y)\right)^2}{R_2(\theta) - R_1(\theta) + \frac{1}{4}}.$$

This bound is tighter than the second order one if the disagreement $\mathbb{E}_{h,h'} \mathbb{E}_\mathcal{P}(\mathbb{1}(h(x) \neq h'(X)))$ is greater than $R_2(\theta)$ [Masegosa et al., 2020].

From a practical point of view, one could minimize the generalization bounds of one of the above methods to learn a weight distribution over an ensemble of predictors. However, this could lead to sub-optimal MV classifiers. To illustrate this behavior we plot in Figure 3 the decision surfaces learnt by the minimization of a PAC-Bayesian bound on each of the aforementioned oracle bounds. These plots provide evidence that, when the base classifiers are weak, state-of-the-art PAC-Bayesian methods do not necessarily build powerful ensembles (failing to improve upon a Naive Bayes approach [Berend and Kontorovich, 2015]). *First Order* concentrates on few base classifiers, as previously observed by Lorenzen et al. [2019] and Masegosa et al. [2020], while *Second Order* and *C-Bound* fail to leverage the diversity in the whole set of classifiers. Indeed, in this setting the base classifiers are weak, but diverse enough so that there exists an optimal combination of them that perfectly splits the two classes without error. However, the optimization of the PAC-Bayes guarantees over *Second Order* and *C-Bound* are shown to select a small subset of base classifiers which is not enough to achieve good performance. On the contrary, *Binomial* is able to fit the problem by drawing more than 2 voters, but it provides generalization bounds that are loose even when the learned model exhibits good generalization capabilities, as in this case.

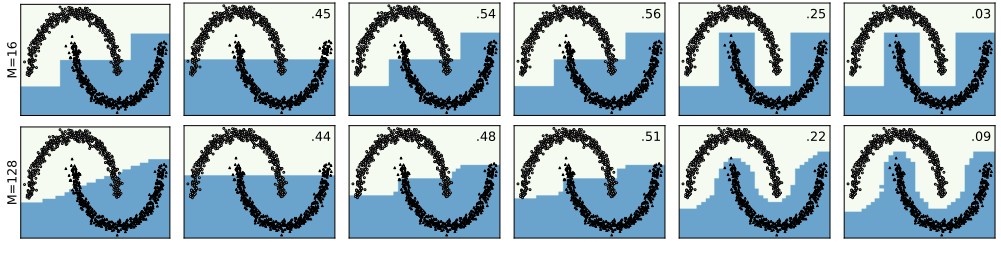

(a) Bayesian NB   (b) First Order  (c) Second Order  (d) C-Bound   (e) Binomial    (f) ours

Figure 3: Decision surface for *Bayesian Naive Bayes*, the PAC-Bayesian methods *First Order*, *Second Order*, *C-Bound* and *Binomial* (with $N = 100$) and our method on the *two-moons* dataset (where each half-circle is a class and inputs lie in $\mathcal{X} = [-2, 2]^2$) with 16 (top) and 128 (bottom) decision stumps as base classifiers (axis-aligned and evenly distributed over the input space). Predicted labels are plotted with different colors, and training points are marked in black. When available, the value of the generalization bound is marked in the right-hand-side-top corner.

## 3   Our approach: stochastic weighted majority vote

In our framework for deriving generalization bound for Majority Vote classifiers, we consider $f_\theta$ as a realization of a distribution of majority vote classifiers, with parameter $\theta \in \Theta \subseteq \mathbb{R}^M$ and probability measure $\rho$, as represented in Figure 2. The main advantage of considering a stochastic majority vote is that it allows to derive and optimize PAC-Bayesian generalization bounds directly on its true risk and to fully leverage the set of base classifiers. The true risk of the proposed stochastic weighted majority vote takes into account the whole distribution of MVs $\rho(\theta)$, as follows:

$$\int_\Theta R(f_\theta)\rho(d\theta) = \int_\Theta \mathbb{E}_\mathcal{P}\, \mathbb{1}(W_\theta(X, Y) \geqslant 0.5)\rho(d\theta) = \mathbb{E}_\mathcal{P} \int_\Theta \mathbb{1}(W_\theta(X, Y) \geqslant 0.5)\rho(d\theta). \quad (4)$$

Given its stochastic nature, in order to evaluate and/or minimize Equation (4) we can either (i) compute its closed form or (ii) approximate it (*e.g.* through Monte Carlo methods). In both cases, assumptions have to be made on the form of $\rho$. As the components of $\theta$ are constrained to sum to one, $\theta$ lies in the $(M\text{-}1)$-simplex: $\Theta = \Delta^{(M\text{-}1)}$, hence natural choices for its probability measure are *e.g.* the Dirichlet or the Logit Normal distributions. In the following, we show that under Dirichlet assumptions, we can derive an analytical and differentiable form of the risk of a stochastic MV.

### 3.1   Exact risk under Dirichlet assumptions

First of all, we recall that the probability density function of the Dirichlet distribution is defined by:

$$\theta \sim \mathcal{D}(\alpha_1, \ldots, \alpha_M), \quad \rho(\theta) = \frac{1}{B(\alpha)} \prod_{j=1}^M (\theta_j)^{\alpha_j - 1}, \quad (5)$$

with $\alpha = [\alpha_j \in \mathbb{R}^+]_{j=1}^M$ the vector of concentration parameters and $B(\alpha)$ a normalization factor (see App A for its definition). Notice that by taking $\alpha$ as the vector of all ones, the distribution corresponds to a uniform distribution over the $(M\text{-}1)$-simplex $\Delta^{M-1}$.

Under these assumptions for the MV distribution, a closed form can be derived for the expected risk:

**Lemma 1.** *For a given $(x, y) \sim \mathcal{P}$, let $w = \{j | h_j(x) \neq y\}$ be the set of indices of the base classifiers that misclassify $(x, y)$ and $c = \{j | h_j(x) = y\}$ be the set of indices of the base classifiers that correctly classify $(x, y)$. The expected error (or $01$-loss) for $(x, y)$ of the stochastic majority vote under $\theta \sim \mathcal{D}(\alpha_1, \ldots, \alpha_M)$ is equal to*

$$\int_\Theta \mathbb{1}(W_\theta(x, y) \geqslant 0.5)\rho(d\theta) = I_{0.5}\left(\sum_{j \in c} \alpha_j, \sum_{j \in w} \alpha_j\right), \quad (6)$$

*with $I_{0.5}(\cdot)$ the regularized incomplete beta function evaluated at $0.5$.*

*Proof.* We rewrite $W_\theta$ as $W_\theta(x,y) = \sum_{j=1}^{M} \theta_j \mathbb{1}(h_j(x) \neq y) = \sum_{j \in w} \theta_j$, and use the aggregation property of Dirichlet distributions to show that $W_\theta$ follows a bivariate Dirichlet distribution (*a.k.a.* Beta distribution):

**Lemma 2.** *If $\theta \sim \mathcal{D}(\alpha_1, \ldots, \alpha_M)$, then for any $j \in [M]$ and $j' \in [1, M] \setminus \{j\}$ the variable $\theta'$ formed by dropping $\theta_j$ and $\theta_{j'}$ and adding their sum also follows a Dirichlet distribution*

$$(\theta_1, \ldots, \theta_M, \theta_j + \theta_{j'}) \sim \mathcal{D}(\alpha_1, \ldots, \alpha_M, \alpha_j + \alpha_{j'}).$$

A proof of this property can be found in `https://vannevar.ece.uw.edu/techsite/papers/documents/UWEETR-2010-0006.pdf`. Hence $W_\theta$ follows a Beta distribution over the two sets of wrong and correct base classifiers:

$$\mathbb{P}[W_\theta(x,y) = \omega] = \mathbb{P}\left[\sum_{j \in w} \theta_j = \omega\right] \mathbb{P}\left[\sum_{j \in c} \theta_j = 1 - \omega\right]$$

$$\implies W_\theta(x,y) \sim \mathcal{D}\left(\sum_{j \in w} \alpha_j, \sum_{j \in c} \alpha_j\right) \quad \text{by aggregation.} \tag{7}$$

Finally, notice that the expected error is related to the cumulative probability function of $W_\theta$, the incomplete beta function $I_p : \mathbb{R}^+ \times \mathbb{R}^+ \to [0, 1]$:

$$\int_\Theta \mathbb{1}(W_\theta(x,y) \geqslant 0.5)\rho(d\theta) = \int_{0.5}^{1} \mathbb{P}[dW_\theta(x,y)] \tag{8}$$

$$= 1 - I_{0.5}\left(\sum_{j \in w} \alpha_j, \sum_{j \in c} \alpha_j\right) = I_{0.5}\left(\sum_{j \in c} \alpha_j, \sum_{j \in w} \alpha_j\right). \tag{9}$$

Equation (9) follows by symmetry of the incomplete beta function: $I_p(a, b) = 1 - I_{1-p}(b, a)$. $\quad\square$

The expected risk $R(\rho)$ can be then expressed as follows:

$$R(\rho) = \mathbb{E}_{\mathcal{P}} \int_\Theta \mathbb{1}(W_\theta(x,y) \geqslant 0.5)\rho(d\theta) = \mathbb{E}_{\mathcal{P}} I_{0.5}\left(\sum_{j \in c} \alpha_j, \sum_{j \in w} \alpha_j\right). \tag{10}$$

Importantly, this exact form of the risk is differentiable, hence can be directly optimized by gradient-based methods.

### 3.2 Monte Carlo approximated risk

We now propose a relaxed Monte Carlo (MC) optimization scheme for those distributions that do not admit an analytical form of the expected risk, unlike the Dirichlet one. This second strategy is also suited to speed up training, in some cases, as the derivatives of the exact risk depend on the hyper-geometric function and can be slow to evaluate (see App. A.3). With the approximated scheme, in order to update $\alpha$ by gradient descent we need to relax the true risk as the gradients of the 01-loss are always null for discrete $W_\theta$. In practice, we make use of a *tempered* sigmoid loss $\sigma_c(x) = \frac{1}{1+\exp(-cx)}$ with slope parameter $c \in \mathbb{R}^+$. De facto this corresponds to solving a relaxation of the problem and not its exact form [Nesterov, 2005]. At each iteration of the MC optimization algorithm we perform:

1. Draw a sample $\{\theta_t \sim \rho(\alpha)\}_{t=1}^{T}$ using the implicit reparameterization trick [Figurnov et al., 2018, Jankowiak and Obermeyer, 2018];

2. Compute the relaxed empirical risk $\sum_{t=1}^{T} \hat{R}_{\sigma_c}(\theta) = \sum_{t=1}^{T} \sum_{i=1}^{n} \sigma_c(W_{\theta_t}(x_i, y_i) - 0.5)$;

3. Update $\alpha$ by gradient descent.

Notice that when considering Dirichlet distributions for the posterior and the prior, at inference time the empirical PAC-Bayesian bounds can still be evaluated using the exact form of Lemma 1.

A drawback of the approximated scheme is that it has a complexity linear in the number of MC draws $T$, but also linear in the number of predictors $M$, as sampling over the simplex requires sampling from $O(M)$ distributions, one per base classifier. As an example, sampling from a Dirichlet over the $(M\text{-}1)$-simplex is usually implemented as sampling from $M$ Gamma distributions and normalizing the samples so that they lie on the simplex. In contrast, the exact formulation's complexity is constant in $M$ as it depends only on the sets of wrong and of correct predictors, hence on a constant number of variables (2) no matter the number of predictors $M$. In Section 5.1 we empirically study the trade-off between training time and accuracy, showing in which regimes it is more convenient to optimize the relaxed MC risk than optimizing the exact one, and viceversa.

## 4  PAC-Bayesian generalization guarantees

We now derive PAC-Bayesian generalization upper bounds for the proposed stochastic MV. In our context, upper bounds can be derived for studying the gap between true and empirical risk considering a prior distribution $\pi$ over the hypothesis space $\Theta$. In this paper, we make use of one of the tightest classical PAC-Bayesian bound [Seeger, 2002, Maurer, 2004]:

**Theorem 1** (Seeger's bound). *For any $\pi$ over $\Theta$ and $\delta \in (0,1)$ with probability at least $1-\delta$ over samples $S = \{(x_i, y_i) \sim \mathcal{P}\}_{i=1}^n$ of size $n$ we have simultaneously for any posterior $\rho$ over $\Theta$:*

$$\int_\Theta R(f_\theta)\rho(d\theta) \leqslant \mathrm{kl}^{-1}\left(\int_\Theta \hat{R}(f_\theta)\rho(d\theta) \, , \, \frac{\mathrm{KL}(\rho,\pi) + \ln\left(\frac{2\sqrt{n}}{\delta}\right)}{n}\right), \tag{11}$$

*with $\hat{R}(f_\theta) = \frac{1}{n}\sum_{i=1}^n \mathbb{1}(W_\theta(x_i, y_i) \geqslant 0.5)$ the empirical risk on sample $S$, $\mathrm{KL}(\rho,\pi) = \int_\Theta \rho(\theta) \log \frac{\rho(\theta)}{\pi(\theta)} \, d\theta$ the KL divergence and $\mathrm{kl}^{-1}(q, \epsilon)$ the inverse of the binary KL divergence defined as $\mathrm{kl}^{-1}(q, \epsilon) = \max\{p \in [0,1] \mid \mathrm{kl}(q, p) \leqslant \epsilon\}$.*

A proof of Theorem 1 can be found in Seeger [2002]. The $\mathrm{kl}^{-1}$ function can be evaluated via the bisection method and optimized by gradient descent, as proposed in Reeb et al. [2018]. Note that our contributions do not restrict the choice of generalization bound.

Importantly, Theorem 1 is valid when the prior $\pi$ is independent from the data. Thus it cannot be evaluated with base classifiers learned from the training sample. However, it is known that considering a data-dependent prior can lead to tighter PAC-Bayes bounds [Dziugaite et al., 2021]. Following recent works on PAC-Bayesian bounds with data-dependent priors [Thiemann et al., 2017, Mhammedi et al., 2019], we derive a cross-bounding certificate that allows us to learn and evaluate the set of base classifiers without held-out data. More precisely, we split the training data $S$ into two subsets ($S_{\leqslant m}=\{(x_i, y_i) \in S\}_{i=1}^m$ and $S_{>m}=\{(x_i, y_i) \in S\}_{i=m+1}^n$) and we learn a set of base classifiers on each data split independently (determining the hypothesis spaces $\Theta_{\leqslant m}$ and $\Theta_{>m}$). We refer to the prior distribution over $\Theta_{\leqslant m}$ as $\pi_{\leqslant m}$ and to the prior distribution over $\Theta_{>m}$ as $\pi_{>m}$. In the same way, we can then define a posterior distribution per hypothesis space: $\rho_{\leqslant m}$ and $\rho_{>m}$. The following theorem shows that we can bound the expected risk of any convex combination of the two posteriors, as long as their empirical risks are evaluated on the data split that was not used for learning their respective priors.

**Theorem 2** (Seeger's bound with informed priors). *Let $\pi_{\leqslant m}$ and $\rho_{\leqslant m}$ be the prior and posterior distributions on $\Theta_{\leqslant m}$, and $\pi_{>m}$ and $\rho_{>m}$ the prior and posterior distributions on $\Theta_{>m}$. For any $p \in (0,1)$ and $\delta \in (0,1)$ with probability at least $1-\delta$ over samples $S = \{(x_i, y_i) \sim \mathcal{P}\}_{i=1}^n$ we have*

$$\mathrm{kl}\left(p\hat{R}(\rho_{>m}) + (1-p)\hat{R}(\rho_{\leqslant m}) \middle\| pR_{\leqslant m}(\rho_{>m}) + (1-p)_{>m}(\rho_{\leqslant m})\right)$$

$$\leqslant \frac{p \, KL(\rho_{>m}, \pi_{>m})}{m} + \frac{(1-p) \, KL(\rho_{\leqslant m}, \pi_{\leqslant m})}{n-m} + \frac{\ln \frac{4\sqrt{m(n-m)}}{\delta}}{n},$$

*with $R(\rho_{>m}) = \int_{\Theta_{>m}} R(f_\theta)\rho(d\theta)$, and $R(\rho_{\leqslant m}) = \int_{\Theta_{\leqslant m}} R(f_\theta)\rho(d\theta)$, and $\hat{R}_{\leqslant m}(\rho_{>m}) = \int_{\Theta_{>m}} \hat{R}(f_\theta)\rho(d\theta)$, and $\hat{R}_{\leqslant m}(\rho_{>m}) = \int_{\Theta_{\leqslant m}} \hat{R}(f_\theta)\rho(d\theta)$.*

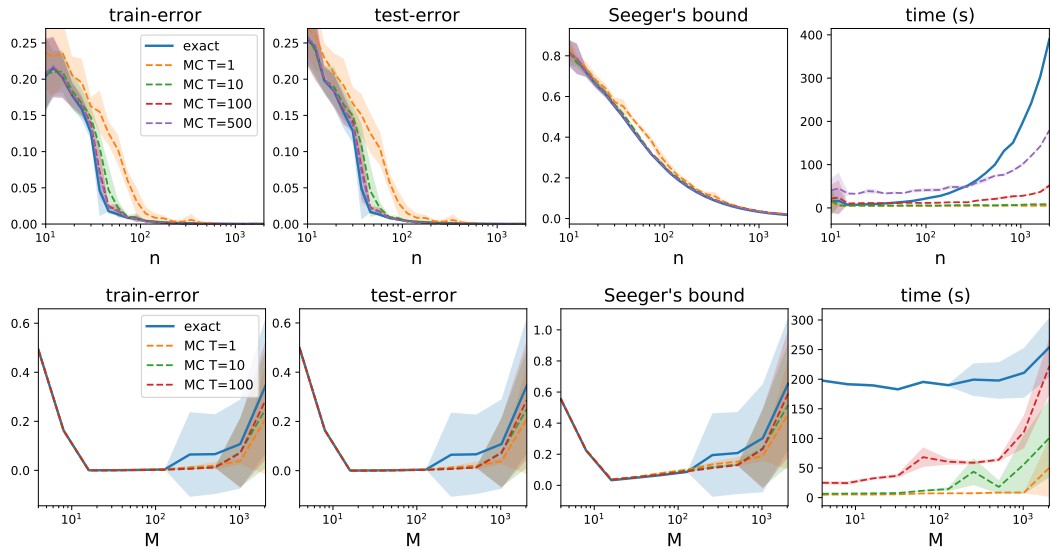

Figure 4: Average performance for 10 trials of *exact* and *MC* variants of our method as a function of the number of training points $n$ (top, $M$ fixed to 16) or of decision stumps $M$ (bottom, $n = 1000$), both represented in logarithmic scale, for different number of MC draws $T$.

The result follows by kl's convexity. The complete proof is reported in App. B. In practice, following Mhammedi et al. [2019] we set $m = \frac{n}{2}$ and $p = \frac{m}{n}$, and we learn the base classifiers by empirical risk minimization.

**Comparison with existing bounds.**    Until now, we considered bounds for the expected risk over the space of MVs but not for a single realization $f_\theta$, unlike state-of-the-art methods. Under Dirichlet assumptions, we can bound the risk of the expected MV $f_{\hat{\theta}}$ by twice the expected risk (see App A.4): $R(f_{\hat{\theta}}) = R(\mathbb{E}_{\rho(\theta)} f_\theta) \leqslant 2\mathbb{E}_{\rho(\theta)} R(f_\theta)$. The obtained oracle bound would comprise an irreducible factor, but its KL term would not degrade, unlike state-of-the-art bounds that account for voter correlation [Lacasse et al., 2006, 2010, Masegosa et al., 2020]. Indeed, the empirical bound does not introduce additional factors on the KL term, such as *Second Order* which has a 2 $KL()$ term and *Binomial* which has a $N$ $KL()$ term. The empirical bound on the deterministic MV could be also refined leveraging works on the disintegration of PAC-Bayesian guarantees [Blanchard and Fleuret, 2007, Catoni, 2007, Rivasplata et al., 2020, Viallard et al., 2021a].

A downside of our method is that the complexity of the KL term grows with the number of base classifiers $M$, unlike the KL on categoricals that tends to 0 in its limit. This results in making the generalization bounds increasingly looser and conservative with growing $M$, even for low empirical risks. From a generalization perspective, our guarantees are hence able to reflect the complexity of the model, expressed as the size of the hypothesis space $M$. However, our risk certificates do not account for redundancy in the voter set. For instance, they are not able to distinguish scenarios where base classifiers are highly correlated (hence less complex hypothesis space) from scenarios where base classifiers are independent (more complex). An expedient for ensuring that generalization bounds are tight consists in learning the hypothesis space: the number of base predictors can then be limited without degrading the performance of the majority vote.

## 5 Experiments

In this section, we empirically evaluate STOCMV, and we compare its generalization bounds and test errors to those obtained with PAC-Bayesian methods learning majority votes. We show that our method allows to derive generalization bounds that are consistently tight (*i.e.* close to the test errors) and non-vacuous (*i.e.* smaller than 1) both when studying ensembles of weak predictors and when studying ensembles of strong ones.

In the following, we consider Dirichlet distributions for the prior and the posterior of our method, and refer to the model obtained by optimizing the exact risk as *exact* and the one obtained by optimizing the approximated one as *MC*. We consider as baselines the PAC-Bayesian methods described in Section 2.1: We refer to the methods optimizing the First Order, Second Order and Binomial empirical bound as *FO*, *SO* and *Bin* respectively. We do not compare with the C-Bound, as it is hard to optimize on large-scale datasets and existing algorithms are suited only for binary classification. All generalization bounds are evaluated with a probability $1-\delta=0.95$ and all prior distributions are set to the uniform (we provide a study for different priors in App. C.2). The posterior parameters ($\alpha$ for our method, $\theta$ for the others) are initialized uniformly in $[0.01, 2]$ (and normalized to sum to 1 for *SO*, *FO* and *Bin*). Finally, for *MC* the sigmoid's slope parameter $c$ is set to 100 and for *Bin* the number of voters drawn at each iteration is set to $N=100$. Code, available at `https://github.com/vzantedeschi/StocMV`, was implemented in pytorch [Paszke et al., 2019] and all experiments were run on a virtual machine with 8 vCPUs and $128Gb$ of RAM.

## 5.1 Comparison of exact and MC variants

For this set of experiments, we optimize Seeger's Bound (Equation (1)) by (batch) Gradient Descent, for $1,000$ iterations and with learning rate equal to 0.1. We study the performance of our method on the binary classification **two-moons** dataset, with 2 features, 2 classes and $\mathcal{N}(0, 0.05)$ Gaussian noise, for which we draw $n$ points for training, and $1,000$ points for testing. Figure 4 reports a comparison of *exact* and *MC* variants in terms of error, generalization bound and training time (in seconds). Increasing the number of MC draws $T$ unsurprisingly allows to recover *exact*'s performance, and at lower computational cost for reasonable values of $M$ and $T$. In general, as *MC* is easily parallelizable, its training time has better dependence on $n$ than *exact*'s one, however it increases with $M$ at a worse rate. When the training sample is large enough (here for $n > 10^2$), *MC* achieves *exact*'s errors and bounds even for $T = 1$. We also observe that the error rates and bounds gradually degrade for higher values of $M$ for both methods. This is due to the KL term increasing with $M$, as highlighted in Section 4, becoming a too strong regularization during training and making the bound looser.

## 5.2 Experiments on real benchmarks

We now compare the considered methods on real datasets and on two different scenarios, depending on the type of PAC-Bayesian bounds that are evaluated: When making use of *data-independent priors*, we chose as voters axis-aligned decision stumps, with thresholds evenly spread over the input space (10 per feature); When making use of *data-dependent priors*, we build Random Forests [Breiman, 2001] as set of voters, each with $M=100$ trees learned bagging $\frac{n}{2}$ points and sampling $\sqrt{d}$ random features to ensure voter diversity, optimizing Gini impurity score and, unless stated otherwise, without bounding their maximal depth.

We consider several classification datasets from UCI [Dua and Graff, 2017], LIBSVM[2] and Zalando [Xiao et al., 2017], of different number of features and of instances. Their descriptions and details on any pre-processing are provided in App. C.2. We train the models by Stochastic Gradient Descent (SGD) using Adam [Kingma and Ba, 2015] with $(0.9, 0.999)$ running average coefficients, batch size equal to $1024$ and learning rate equal to $0.1$ with a scheduler reducing this parameter of a factor of 10 with 2 epochs patience. We fix the maximal number of epochs to 100 and patience equal to 25 for early stopping, and for *MC* we fix $T = 10$ to increase randomness.

We report the test errors and generalization bounds in Figure 5 (additional results are reported in the appendix, in Tables 1 and 2 and Figure 14): We compare the different methods on binary datasets and with data-independent priors in Figure 5a, and on multi-class datasets and with data-dependent priors in Figure 5b. First we notice that the bounds obtained by our method are consistently non vacuous and tighter than those obtained by the baselines on all datasets. Regarding the error rates, our method's performance is generally aligned with the baselines, while it achieved error rates significantly lower than *FO* and *SO* on the perfectly separable *two-moons* dataset. Sensitivity to noise could explain why our method does not outperform the baselines on the studied real problems, as these usually present label and input noise. Indeed our learning algorithm optimizes the 01-loss, which does not distinguish points with margins close or far from $0.5$ because of its discontinuity in $W_\theta = 0.5$. Preliminary results reported in App C.1 seem to confirm this supposition.

---

[2]`https://www.csie.ntu.edu.tw/~cjlin/libsvm/`

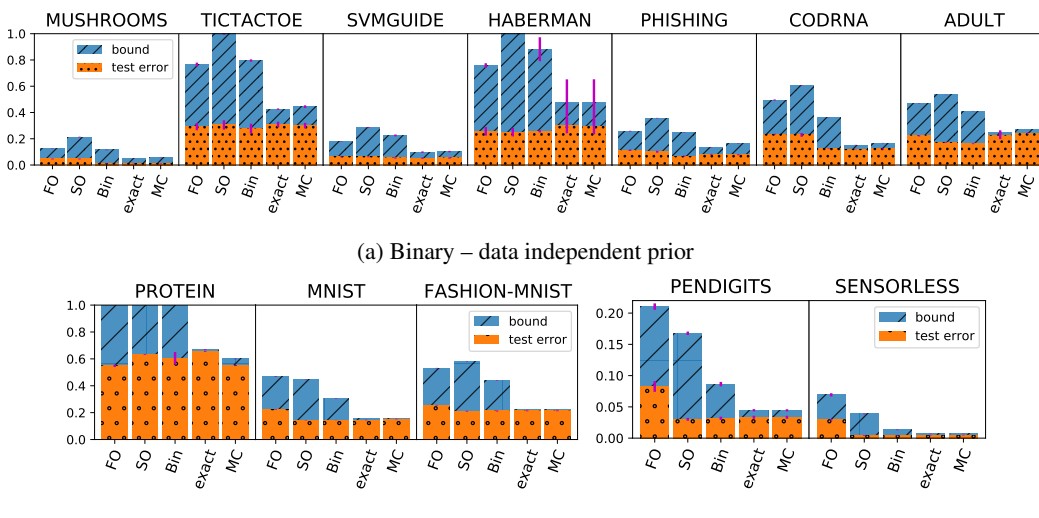

(a) Binary – data independent prior

(b) Multi-class – data dependent prior

Figure 5: Comparison in terms of test error rates and PAC-Bayesian bound values. We report the means (bars) and standard deviations (vertical, magenta lines) over 10 different runs.

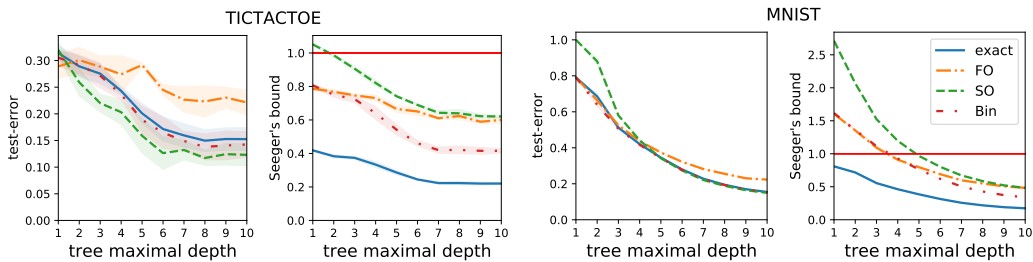

Figure 6: Test error rates and PAC-Bayesian bound values as a function of the maximal depth of the voters (decision trees here). We report the means and standard deviations over 4 different runs, and mark with a red horizontal line the threshold above which Seeger's bounds are vacuous. Additional results are available in the appendix.

Finally, to gain a better understanding of the relation between base classifier strength and performance of the models obtained with the different methods, we further study their performance and generalization guarantees with varying voter strength. As hypothesis set, we learn Random Forests with 200 decision trees in total, as before, but for this experiment we bound their maximal depth between 1 and 10. Constraining the tree depth allows to indirectly control how well the voters fit the training set (as shown in Appendix, Figure 13). Following Lorenzen et al. [2019], we assess the voter strength by computing the expected accuracy of a random voter. All methods' results improve overall when voters get stronger, even though *FO* at a slower pace. Notice that on the considered datasets *SO* is the most sensitive method, particularly suffering from weak base predictors. Our method generally provides test errors comparable with the best baselines and consistently tighter bounds.

## 6 Future work

We propose a stochastic version of the classical majority vote classifier, and we directly analyze and optimize its expected risk through the PAC-Bayesian framework. The benefits on the model accuracy of this direct optimization are however reduced in presence of input noise, and fostering robustness in noisy contexts is the subject of future work. Another potential improvement would consist in tackling the discussed looseness of our generalization bounds with increasing number of base predictors, by accounting for redundancy in the hypothesis space.

## Acknowledgments and Disclosure of Funding

We thank the anonymous reviewers for their constructive feedback and support. This work was partially funded by the French Project APRIORI ANR-18-CE23-0015. Experiments presented in this paper were carried out using the Grid'5000 testbed, supported by a scientific interest group hosted by Inria and including CNRS, RENATER and several Universities as well as other organizations (see `https://www.grid5000.fr`). Pascal Germain is supported by the Canada CIFAR AI Chair Program, and the NSERC Discovery grant RGPIN-2020-07223. Benjamin Guedj acknowledges partial support by the U.S. Army Research Laboratory and the U.S. Army Research Office, and by the U.K. Ministry of Defence and the U.K. Engineering and Physical Sciences Research Council (EPSRC) under grant number EP/R013616/1. Benjamin Guedj acknowledges partial support from the French National Agency for Research, grants ANR-18-CE40-0016-01 and ANR-18-CE23-0015-02.

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
