# A Derivation details under Dirichlet assumptions

## A.1 Dirichlet distribution

The probability density function of the Dirichlet distribution is defined as follows:

$$\theta \sim \mathcal{D}(\alpha_1, \ldots, \alpha_M), \quad \rho(\theta) = \frac{1}{B(\alpha)} \prod_{j=1}^{M} (\theta_j)^{\alpha_j - 1}, \tag{12}$$

with $\alpha = [\alpha_j \in \mathbb{R}^+]_{j=1}^{M}$ the vector of the distribution parameters and $B(\alpha)$ the normalized multivariate beta function, defined using the gamma function $\Gamma$:

$$B(\alpha) = \frac{\prod_{j=1}^{M} \Gamma(\alpha_j)}{\Gamma(\sum_{j=1}^{M} \alpha_j)}, \qquad \Gamma(a) = \int_0^{\infty} t^{a-1} e^{-t} dt. \tag{13}$$

## A.2 KL divergence between Dirichlet distributions

Let $\rho(\theta) = \mathcal{D}(\alpha)$ and $\pi(\theta) = \mathcal{D}(\beta)$, with $\alpha_0 = \sum_{j=1}^{M} \alpha_j$ and $\beta_0 = \sum_{j=1}^{M} \beta_j$. The KL divergence between $\rho$ and $\pi$ is equal to:

$$KL(\rho, \pi) = \ln(\Gamma(\alpha_0)) - \sum_{j=1}^{M} \ln(\Gamma(\alpha_j)) - \ln(\Gamma(\beta_0)) + \sum_{j=1}^{M} \ln(\Gamma(\beta_j)) + \sum_{j=1}^{M} (\alpha_j - \beta_j)(\psi(\alpha_j) - \psi(\alpha_0)) \tag{14}$$

with $\psi(a)$ the digamma function: the first derivative of $\ln(\Gamma(a))$ (see Figure 7).

*Proof.* We have

$$KL(\rho, \pi) = \int_{\Theta} \rho(\theta) \ln \left( \frac{\rho(\theta)}{\pi(\theta)} \right) d\theta \tag{15}$$

$$= \int_{\Theta} \ln \left( \frac{B(\beta) \prod_{j=1}^{M} \theta_j^{\alpha_j - 1}}{B(\alpha) \prod_{j=1}^{M} \theta_j^{\beta_j - 1}} \right) \rho(\theta) d\theta \tag{16}$$

$$= \int_{\Theta} \left( \ln B(\beta) - \ln B(\alpha) + \ln \left( \prod_{j=1}^{M} \theta_j^{\alpha_j - \beta_j} \right) \right) \rho(\theta) d\theta \tag{17}$$

$$= \int_{\Theta} \left( \ln B(\beta) - \ln B(\alpha) + \sum_{j=1}^{M} (\alpha_j - \beta_j) \ln \theta_j \right) \rho(\theta) d\theta \tag{18}$$

$$= \ln B(\beta) - \ln B(\alpha) + \sum_{j=1}^{M} (\alpha_j - \beta_j) \int_{\Theta} \ln \theta_j \, \rho(\theta) d\theta \tag{19}$$

$$= \sum_{j=1}^{M} \ln(\Gamma(\beta_j)) - \ln(\Gamma(\beta_0)) + \ln(\Gamma(\alpha_0)) - \sum_{j=1}^{M} \ln(\Gamma(\alpha_j))$$

$$+ \sum_{j=1}^{M} (\alpha_j - \beta_j)(\psi(\alpha_j) - \psi(\alpha_0)). \tag{20}$$

Equation (20) follows by definition of Dirichlet's geometric mean:

$$\int_{\Theta} \ln \theta_j \, \rho(\theta) d\theta = \psi(\alpha_j) - \psi(\alpha_0).$$

$\square$

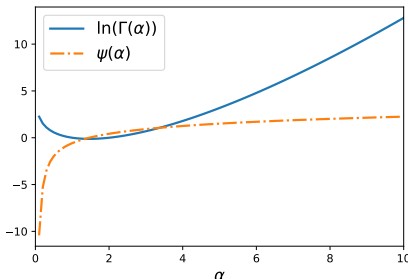

Figure 7: Functions of the gamma family.

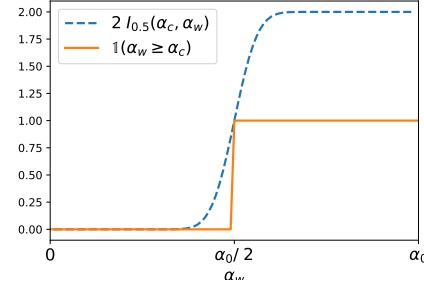

Figure 8: Oracle bound on the expected MV.

### A.3 Partial derivatives for gradient-based optimization

When optimizing our objective functions by gradient descent, we make use of the following partial derivatives *w.r.t.* the Dirichlet parameters $\{\alpha_j\}_{j=1}^M$.

The partial derivatives of the KL divergence are

$$\nabla_{\alpha_j} KL(\rho, \pi) = \psi(\alpha_0) - \psi(\alpha_j) + \psi(\alpha_j) - \psi(\alpha_0) + (\alpha_j - \beta_j)(\psi'(\alpha_j) - \psi'(\alpha_0)) \qquad (21)$$
$$= (\alpha_j - \beta_j)(\psi'(\alpha_j) - \psi'(\alpha_0)). \qquad (22)$$

For a given data point $(x, y) \sim \mathcal{P}$, recall that $w = \{j | h_j(x) \neq y\}$ is the set of indices of the base classifiers that misclassify $(x, y)$ and $c = \{j | h_j(x) = y\}$ is the set of indices of the base classifiers that correctly classify $(x, y)$. Let us define $w(\alpha) = \sum_{j \in w} \alpha_j$ and $c(\alpha) = \sum_{j \in c} \alpha_j = \alpha_0 - w(\alpha)$. For any $j \in c$ we have

$$\nabla_{\alpha_j} I_{0.5}(c(\alpha), w(\alpha)) =$$
$$(\ln 0.5 - \psi(c(\alpha)) + \psi(\alpha_0)) I_{0.5}(c(\alpha), w(\alpha))$$
$$- 0.5^{c(\alpha)} \frac{\Gamma(c(\alpha))\Gamma(\alpha_0)}{\Gamma(w(\alpha))} {}_3F_2(c(\alpha), c(\alpha), 1 - w(\alpha); c(\alpha) + 1, c(\alpha) + 1; 0.5) \qquad (23)$$

and for any $j \in w$ we have

$$\nabla_{\alpha_j} I_{0.5}(c(\alpha), w(\alpha)) =$$
$$(- \ln 0.5 + \psi(w(\alpha)) - \psi(\alpha_0)) I_{0.5}(c(\alpha), w(\alpha))$$
$$+ 0.5^{w(\alpha)} \frac{\Gamma(w(\alpha))\Gamma(\alpha_0)}{\Gamma(c(\alpha))} {}_3F_2(w(\alpha), w(\alpha), 1 - c(\alpha); w(\alpha) + 1, w(\alpha) + 1; 0.5) \qquad (24)$$

where ${}_3F_2(a, b, c; d, e; z)$ is the generalized hyper-geometric function:

$$_3F_2(a, b, c; d, e; z) = \sum_{t=1}^{T} \frac{(a)^t \, (b)^t \, (c)^t \, z^t}{(d)^t \, (e)^t \, t!}$$

with $(.)^t$ the rising factorial.

The hyper-geometric function can be slow to evaluate, as its convergence rate varies depending on $c(\alpha)$ and $w(\alpha)$. A possible strategy for speeding up this evaluation, apart from parallelizing computations, would be dynamic programming, *i.e.* storing the gradients of the incomplete beta function, as it is likely to be evaluated several times for the same $c(\alpha)$ and $w(\alpha)$.

### A.4 Oracle bound on expected Majority Vote

We now derive an oracle bound for the risk of the expected majority vote, given a distribution $\rho(\theta)$ having a Dirichlet form and concentration vector $\alpha = \{\alpha_j\}_{j=1}^M$. The oracle bound can then be used

to derive empirical PAC-Bayesian bounds on the true risk of the expected MV. The expected MV is parameterized by the mean weighting vector $\hat{\theta}$, which for a Dirichlet distribution is given by

$$\hat{\theta} = \mathbb{E}_\rho \, \theta = \frac{\alpha}{\alpha_0} \quad \text{with} \quad \alpha_0 = \sum_{j=1}^M \alpha_j.$$

For a given $(x, y) \sim \mathcal{P}$, let us define $w(x, y) = \{j | h_j(x) \neq y\}$ the set of indices of the base classifiers that misclassify $(x, y)$ and $c(x, y) = \{j | h_j(x) = y\}$ be the set of indices of the base classifiers that correctly classify $(x, y)$. The risk of the expected MV can then be measured as follows:

$$R(f_{\hat{\theta}}) = \mathbb{P}\left[W_{\hat{\theta}} \geqslant 0.5\right] \tag{25}$$

$$= \mathbb{P}\left[\mathbb{E} \, W_\theta \geqslant 0.5\right] \tag{26}$$

$$= \mathbb{P}_{(x,y)\sim\mathcal{P}}\left[\sum_{j|h_j(x)\neq y} \frac{\alpha_j}{\alpha_0} \geqslant 0.5\right] \tag{27}$$

$$= \mathbb{P}_{(x,y)\sim\mathcal{P}}\left[\sum_{j\in w(x,y)} \alpha_j \geqslant \sum_{j\in c(x,y)} \alpha_j\right]. \tag{28}$$

Equation (26) follows by linearity of $W_\theta$.

We recall that the expected risk is given by

$$\mathbb{E}_{(x,y)\sim\mathcal{P}} \, I_{0.5}\left(\sum_{j\in c(x,y)} \alpha_j, \sum_{j\in w(x,y)} \alpha_j\right) = \mathbb{E}_{(x,y)\sim\mathcal{P}} \, I_{0.5}\left(\alpha_0 - \alpha_{w(x,y)}, \alpha_{w(x,y)}\right)$$

with $\alpha_{w(x,y)} = \sum_{j\in w(x,y)} \alpha_j$. As $I_{0.5}()$ is monotonically increasing in its second argument and $I_{0.5}\left(\frac{\alpha_0}{2}, \frac{\alpha_0}{2}\right) = 0.5$, we can then relate the two risks:

$$R(f_{\hat{\theta}}) \leqslant 2\mathbb{E}_\rho R(f_\theta). \tag{29}$$

See Figure 8 for an illustration.

## B    Proof of Theorem 2

Let us define the following empirical risks

$$\hat{R}_{\leqslant m}(\rho_{>m}) = \int_{\Theta_{>m}} \hat{R}_{\leqslant m}(f_\theta)\rho_{>m}(d\theta) \text{ and } \hat{R}_{>m}(\rho_{\leqslant m}) = \int_{\Theta_{\leqslant m}} \hat{R}_{>m}(f_\theta)\rho_{\leqslant m}(d\theta)$$

and true risks

$$R(\rho_{>m}) = \int_{\Theta_{>m}} R(f_\theta)\rho_{>m}(d\theta) \text{ and } R(\rho_{\leqslant m}) = \int_{\Theta_{\leqslant m}} R(f_\theta)\rho_{\leqslant m}(d\theta).$$

**Theorem 2** (Seeger's bound with informed priors)**.** *Let $\pi_{\leqslant m}$ and $\rho_{\leqslant m}$ be the prior and posterior distributions on $\Theta_{\leqslant m}$ and $\pi_{>m}$ and $\rho_{>m}$ the prior and posterior distributions on $\Theta_{>m}$. For any $p \in (0, 1)$ and $\delta \in (0, 1)$ with probability at least $1-\delta$ over samples $S = \{(x_i, y_i)\sim\mathcal{P}\}_{i=1}^n$ of size $n$ we have simultaneously:*

$$\text{kl}\Big(pR(\rho_{>m}) + (1-p)R(\rho_{\leqslant m}) \Big\| p\hat{R}_{\leqslant m}(\rho_{>m}) + (1-p)\hat{R}_{>m}(\rho_{\leqslant m})\Big)$$

$$\leqslant \frac{p \, KL(\rho_{>m}, \pi_{>m})}{m} + \frac{(1-p) \, KL(\rho_{\leqslant m}, \pi_{\leqslant m})}{n-m} + \frac{\ln \frac{4\sqrt{m(n-m)}}{\delta}}{n}$$

*with $R(\rho_{>m}) = \int_{\Theta_{>m}} R(f_\theta)\rho(d\theta)$, and $R(\rho_{\leqslant m}) = \int_{\Theta_{\leqslant m}} R(f_\theta)\rho(d\theta)$, and $\hat{R}_{\leqslant m}(\rho_{>m}) = \int_{\Theta_{>m}} \hat{R}(f_\theta)\rho(d\theta)$, and $\hat{R}_{\leqslant m}(\rho_{>m}) = \int_{\Theta_{\leqslant m}} \hat{R}(f_\theta)\rho(d\theta)$.*

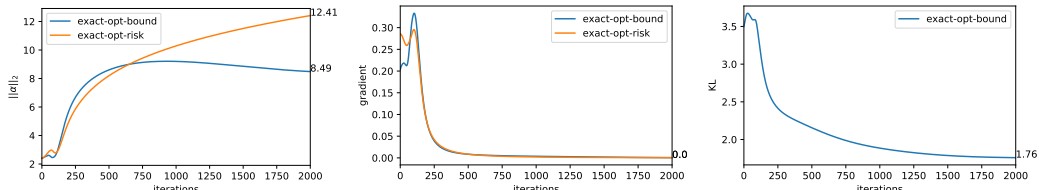

Figure 9: Evolution of $\|\alpha\|_2$ (left), its gradient (center) and $KL(\alpha, \beta)$ (right) during training using the *exact* method. In each plot we compare the values obtained when optimizing Seeger's Bound (Equation (1)) with those obtained when optimizing only the empirical risk.

*Proof.* From the joint convexity of the binary kl divergence, we have for any $p \in [0, 1]$

$$\mathrm{kl}\Big(p\hat{R}(\rho_{>m}) + (1-p)\hat{R}(\rho_{\leqslant m}) \,\Big\|\, pR_{\leqslant m}(\rho_{>m}) + (1-p)R_{>m}(\rho_{\leqslant m})\Big)$$

$$\leqslant p\mathrm{kl}\Big(\hat{R}(\rho_{>m})\Big\|R_{\leqslant m}(\rho_{>m})\Big) + (1-p)\mathrm{kl}\Big(\hat{R}(\rho_{\leqslant m})\Big\|R_{>m}(\rho_{\leqslant m})\Big).$$

For $\forall\rho_{>m}$ defined on $\Theta_{>m}$:

$$\mathbb{P}_{S_{\leqslant m}\sim\mathcal{P}^m}\left[p\,\mathrm{kl}\Big(\hat{R}(\rho_{>m})\Big\|R_{\leqslant m}(\rho_{>m})\Big) \leqslant p\frac{KL(\rho_{>m}, \pi_{>m}) + \ln\frac{4\sqrt{m}}{\delta}}{m}\right] \geqslant 1-\frac{\delta}{2}$$

$$\Rightarrow \mathbb{P}_{S\sim\mathcal{P}^n}\left[p\,\mathrm{kl}\Big(\hat{R}(\rho_{>m})\Big\|R_{\leqslant m}(\rho_{>m})\Big) \leqslant p\frac{KL(\rho_{>m}, \pi_{>m}) + \ln\frac{4\sqrt{m}}{\delta}}{m}\right] \geqslant 1-\frac{\delta}{2}; \qquad (30)$$

and for $\forall\rho_{\leqslant m}$ defined on $\Theta_{\leqslant m}$:

$$\mathbb{P}_{S_{>m}\sim\mathcal{P}^{n-m}}\left[(1-p)\mathrm{kl}\Big(\hat{R}(\rho_{\leqslant m})\Big\|R_{>m}(\rho_{\leqslant m})\Big) \leqslant (1-p)\frac{KL(\rho_{\leqslant m}, \pi_{\leqslant m}) + \ln\frac{4\sqrt{n-m}}{\delta}}{n-m}\right] \geqslant 1-\frac{\delta}{2}$$

$$\Rightarrow \mathbb{P}_{S\sim\mathcal{P}^n}\left[(1-p)\,\mathrm{kl}\Big(\hat{R}(\rho_{\leqslant m})\Big\|R_{>m}(\rho_{\leqslant m})\Big) \leqslant (1-p)\frac{KL(\rho_{\leqslant m}, \pi_{\leqslant m}) + \ln\frac{4\sqrt{n-m}}{\delta}}{n-m}\right] \geqslant 1-\frac{\delta}{2}.$$

$$(31)$$

Combining Equation (30) and Equation (31) using the union bound, we obtain the desired result with $1 - \delta$ probability. □

## C   Additional experimental results

### C.1   Analysis of the role of bound regularization on the posterior

**Synthetic dataset.**   We study the behavior of the method during training on a simple toyset, built from two normal distributions $\mathcal{N}_1([-1, 0], \mathrm{diag}([0.1, 1]))$ and $\mathcal{N}_2([1, 0], \mathrm{diag}([0.1, 1]))$, one per class, so that the two classes are almost perfectly separable on the first dimension. from each class distribution for training, $n$ points are drawn, and $1,000$ points for testing.

**Base predictors and prior.**   We fix the set of base voters to $M{=}4$ axis-aligned decision stumps: 2 per class and centered in 0 on each dimension, so that the problem is well specified as the optimal classifier is in the predictor set. Then, we set the parameters $\beta$ of the prior distribution all to $0.1$ to encourage sparse solutions for the posterior, and we initialize the posterior's parameters by drawing $\alpha_j \sim \mathcal{U}(0.01, 2)$.

**Study of the impact of optimizing the PAC-Bayesian bound.**   In Figure 9, we compare the evolution of $\alpha$ during training for the exact method and $n{=}50$. In each plot we compare the posterior parameters $\alpha$ obtained when optimizing Bound (Equation (1)) with the one obtained when optimizing only the empirical risk (hence, without any regularization). Namely, we study the evolution during

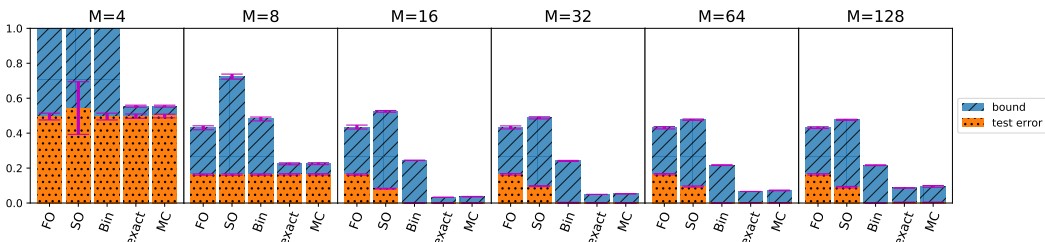

Figure 10: Comparison in terms of test error rates and PAC-Bayesian bound values. Each block corresponds to a different number of predictors $M$. We report the means (bars) and standard deviations (vertical, magenta lines) over 10 different runs.

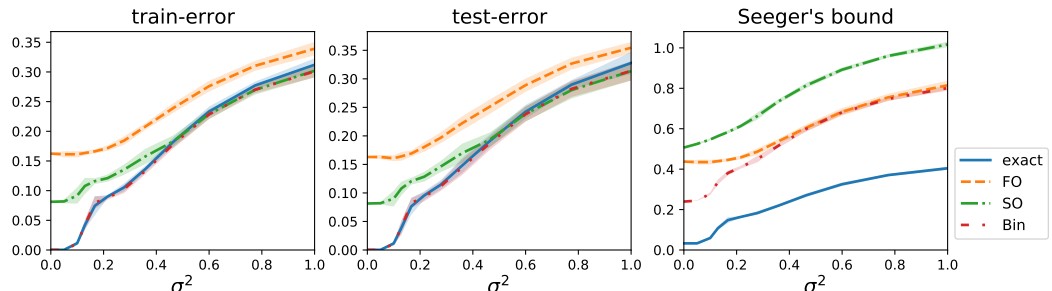

Figure 11: Comparison in terms of error rates and PAC-Bayesian bounds, depending on the magnitude of input noise $\mathcal{N}(0, \sigma^2)$. We report the means and standard deviations over 10 different runs.

training of $\|\alpha\|_2$, of $\alpha$'s gradients and of the KL divergence *w.r.t.* prior: $KL(\rho, \pi)$. We notice that without regularization, $\alpha$ diverges, while it tends to a constant and smaller value when optimizing the bound. This behaviour is not due to optimization instability, as shown by the gradient smoothly tending to 0 for both methods. Instead, it can be explained considering that without regularization the stochastic MV tends to concentrate around the optimal MV, *i.e.* a single $\theta$. This behavior results in reducing the variance of the distribution, thus increasing the concentration parameters $\alpha$. In contrast, when optimizing the bound the model fits the empirical risk and finds optimal solutions that are also close to the prior.

**Comparison with baselines.** We report a comparison of *FO*, *SO*, *Bin* and our method on the binary classification **two-moons** dataset. We report in Figure 10 the results obtained after optimizing the respective PAC-Bayesian bounds. Notice that our bounds are tighter than the baselines, for any value of $M$: this is principally due to the fact that our method consistently obtains lower error rates, being able to better fit the training set (as shown in Figure 3), but also to the fact that our bound is not based on an oracle upper bound on the true risk, unlike the baselines.

**Sensitivity to noise.** We observe that on *two-moons* our method outperforms the baselines also in terms of error rates. However, this does not seem to be the case on real benchmarks where its test errors are generally aligned with those of the baselines. We conduct an additional experiment to study whether this phenomenon is due to sensitivity to noise, such as input noise. Indeed our learning algorithm optimizes the 01-loss, which does not distinguish points with margins close or far from 0.5 because of its discontinuity in $W_\theta = 0.5$. In Figure 11, we assess training and test errors, and Seeger's bound values with increasing input noise. For this experiment, we generated training and test sets as before, and with Gaussian noise $\mathcal{N}(0, \sigma^2)$ added to the inputs. As expected, all methods degrade with increasing noise. In particular, the test errors of *exact* and *Bin* worsen the fastest and the benefits of using them vanish from $\sigma^2 > 0.35$.

## C.2 Additional results on real benchmarks

**Dataset descriptions** We consider several classification datasets from UCI [Dua and Graff, 2017], LIBSVM https://www.csie.ntu.edu.tw/~cjlin/libsvm/ and Zalando [Xiao et al., 2017], of different number of features and of points:

- *Haberman* (UCI): prediction of survival of $n = 306$ patients who had undergone surgery from $d = 3$ anonymized features;

- *TicTacToe* (UCI): determination of a win for player $x$ at TicTacToe game of any of the $n = 958$ board configurations ($d = 9$ categorical states);

- *Svmguide1* (LIBSVM): $d = 4$ features, $n = 7,089$ instances and 2 classes (no description available);

- *Mushrooms* (UCI): prediction of edibility of $n = 8,124$ mushroom sample, given their $d = 22$ categorical features describing their aspect;

- *Phishing* (LIBSVM): prediction of phishing websites ($n = 2456$ websites and $d = 68$ binary encoded features);

- *Adult* (LIBSVM a1a): determining whether a person earns more than 50K a year ($n = 32,561$ people and $d = 123$ binary features);

- *CodRNA* (LIBSVM): detection of non-coding RNAs among $n = 59,535$ instances and from $d = $ features;

- *Pendigits* (UCI): recognition of hand-written digits (10 classes, $d = 9$ features and $n = 12,992$);

- *Protein* (LIBSVM): $d = 357$ features, $n = 24,387$ instances and 3 classes;

- *Shuttle* (UCI): $d = 9$ features, $n = 58,000$ and 7 classes;

- *Sensorless* (LIBSVM): prediction of motor condition ($n = 58,509$ instances and 11 classes), with intact and defective components, from $d = 48$ features extracted from electric current drive signals;

- *MNIST* (LIBSVM): prediction of hand-written digits ($n = 70,000$ instances and 10 classes) from $d = 28 \times 28$ gray-scale images;

- *Fashion-MNIST* (Zalando): prediction of cloth articles ($n = 70,000$ instances and 10 classes) from $d = 28 \times 28$ gray-scale images.

At each run of an algorithm, we randomly split a dataset in training and test sets of sizes $80\% - 20\%$ respectively. Note that we do not make use of a validation set, as we use the PAC-Bayesian bounds as estimate of the test error for model selection. Finally, we convert all categorical features to numerical using an ordinal encoder and z-score all features using the statistics of the training set.

**Choice of prior.** In all the other experiments, we fixed the prior distribution (parameterized by $[\beta_j]_{j=1}^M$) to the uniform, *i.e.* $\beta_j = 1$, $\forall j$. This choice was to make the comparison with the baselines as fair as possible, as their prior was also fixed to the uniform (categorical). However, we can bias the sparsity of the posterior, or conversely its concentration, by choosing a different value for the prior distribution parameters. In some cases, tuning the prior parameters allows to obtain better performance, as reported in Figure 12. In particular, on *Protein* encouraging sparser solutions generally provides better results, confirmed by the fact that the best baseline on this dataset, *FO*, is known to output sparse solutions. On the contrary, on datasets where methods accounting for voter correlation outperform *FO*, such as on *MNIST*, encouraging solutions to be concentrated and close to the simplex mean yields better performance. In general, these results suggest that the choice of prior distribution has a high impact on the learned model's performance and tuning its concentration parameters would be a viable option for improving results.

## C.3 Impact of voter strength

We report the complete study on the impact of voter strength on the learned models. More precisely we provide results for additional datasets as well as the study of the expected strength of a voter as a function of the tree maximal depth. Recall that as hypothesis set, we learn a Random Forest with 100

decision trees for which we bound the maximal depth between 1 and 10. In Figure 13, we can see that limiting the maximal depth is an effective way for controlling the strength of the voters, measured as the expected accuracy of a random voter. Apart from *Protein*, where decision trees do not seem to be a good choice of base predictor, increasing the strength of the voters generally yields more powerful ensembles for all methods. Our method has error rates comparable with the best baselines and enjoys tight and non-vacuous generalization guarantees for any tree depth. Finally, by comparing *SO*'s training and test errors we notice that this method tends to overfit the dataset especially when the base classifiers are weaker (tree depth close to 1).

### C.4 Model entropy and complexity, Summary of main results

We provide two additional elements for assessing the differences between models obtained with our method and models obtained with the upper-bound baselines. Figure 14 reports the values of the entropy of the obtained posteriors, the KL term in the PAC-Bayes bounds, the training error, the test error and the bound value. The first measurement (entropy) assesses the diversity of the obtained posteriors: the higher the entropy, the higher the number of selected base classifiers. The entropy is generally the highest for our models and the lowest for *FO* which has already been shown to select very few base classifiers. The second measurement (KL divergence) is provided to verify that our method obtains tighter generalization guarantees because it does not consist in an upper bound of the

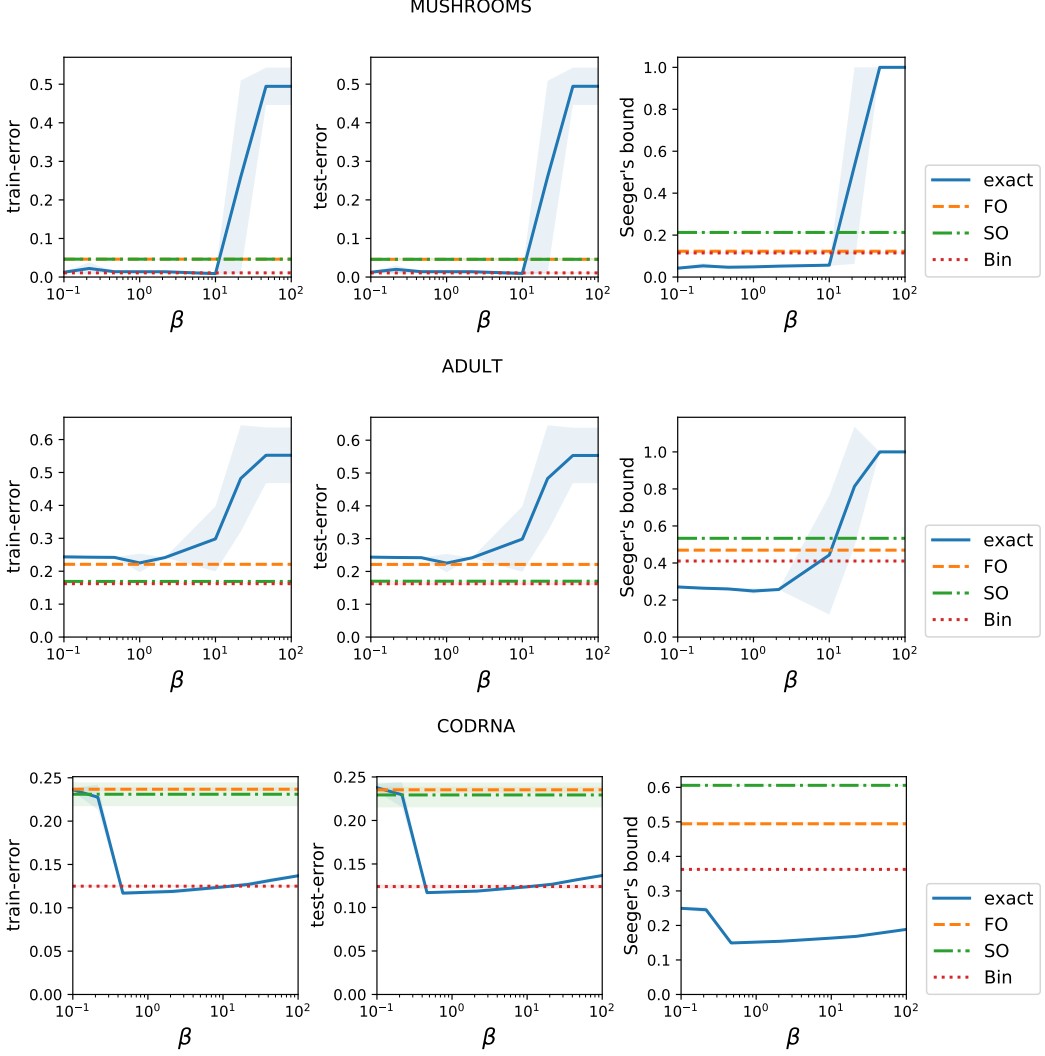

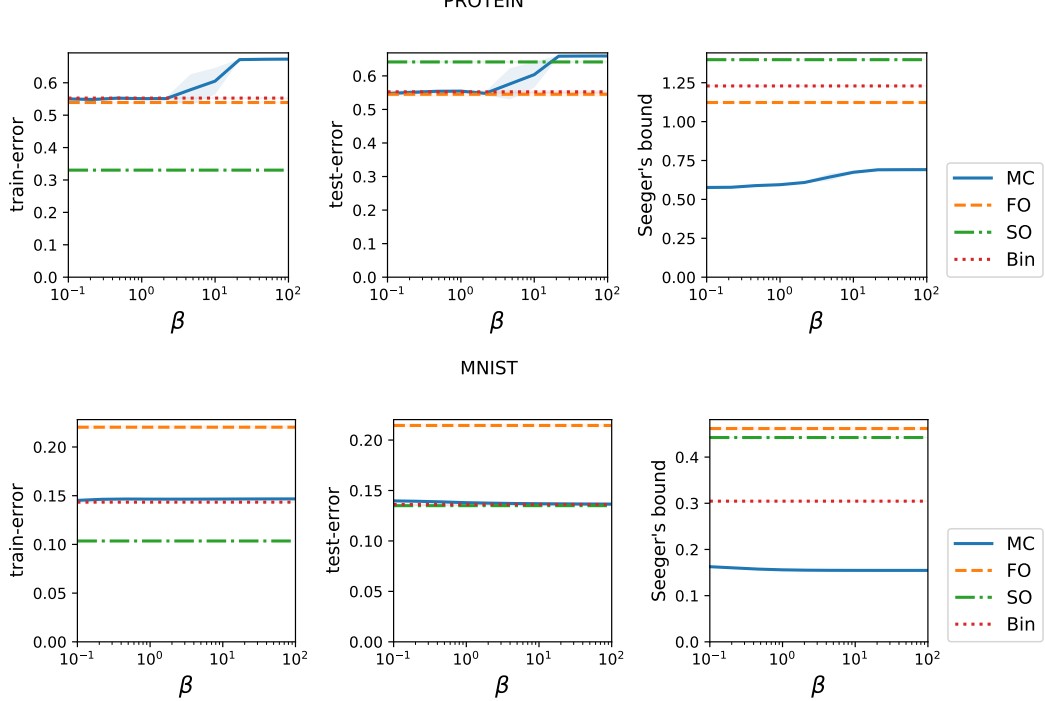

Figure 12: Study of impact of prior on posterior's performance. We fix all $M$ prior's parameters to $\beta$, on the x axis: the smaller $\beta$, the sparser the posterior is encouraged to be. We plot average and standard deviations over 4 trials.

01-loss and not because it obtains models with lower complexity. Indeed, the posteriors optimized with our variants *exact* and *MC* do not necessarily have low KL divergence w.r.t. the prior.

We finally report the detailed comparison on real benchmarks in Figure 14 and Tables 1 and 2.

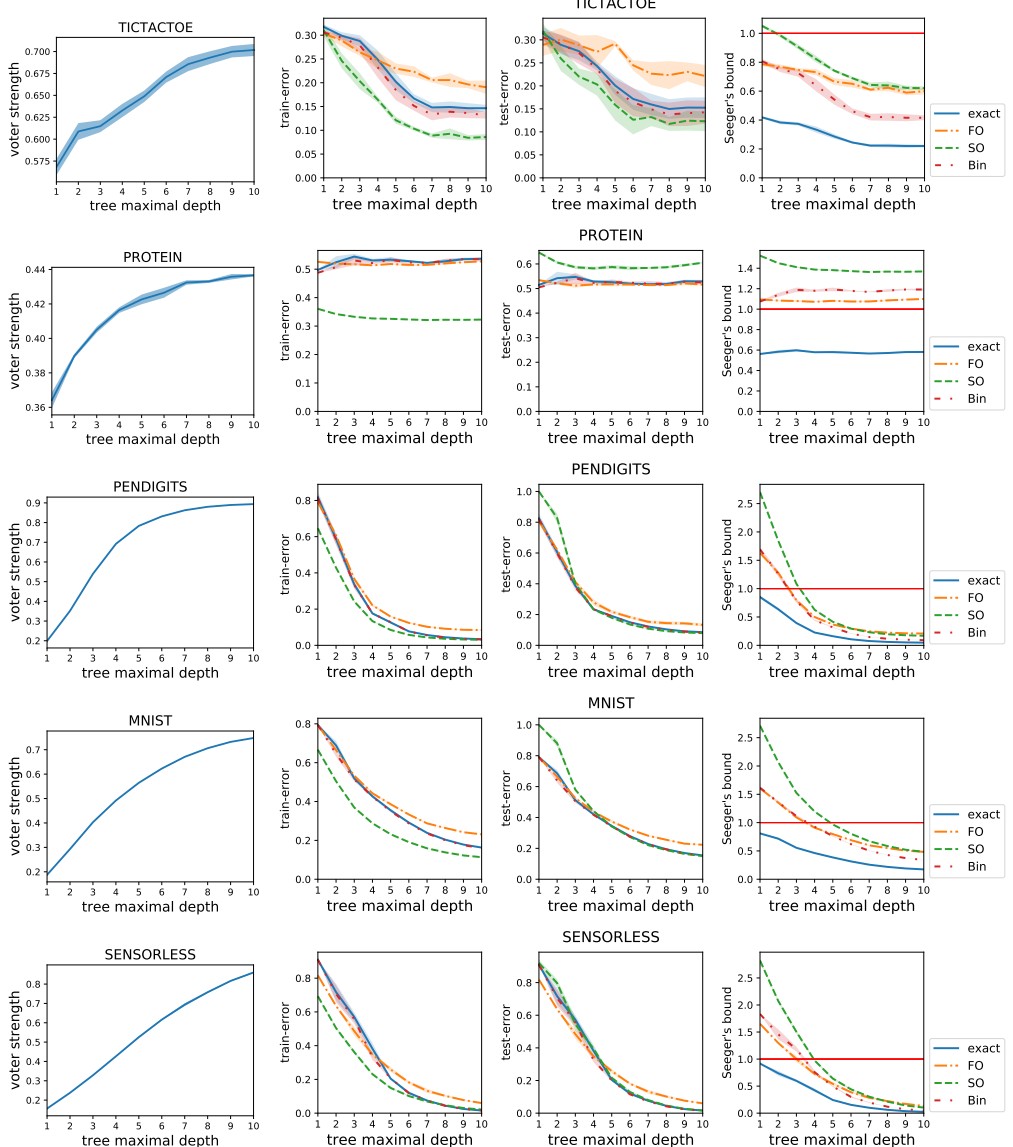

Figure 13: Comparison voter strength (1st column), training error (2nd column), test error (3rd column) and Seeger's bound (4th column) as a function of the tree maximal depth. We mark with a red horizontal line the threshold above which the bounds are vacuous. Results are averaged over 4 trials.

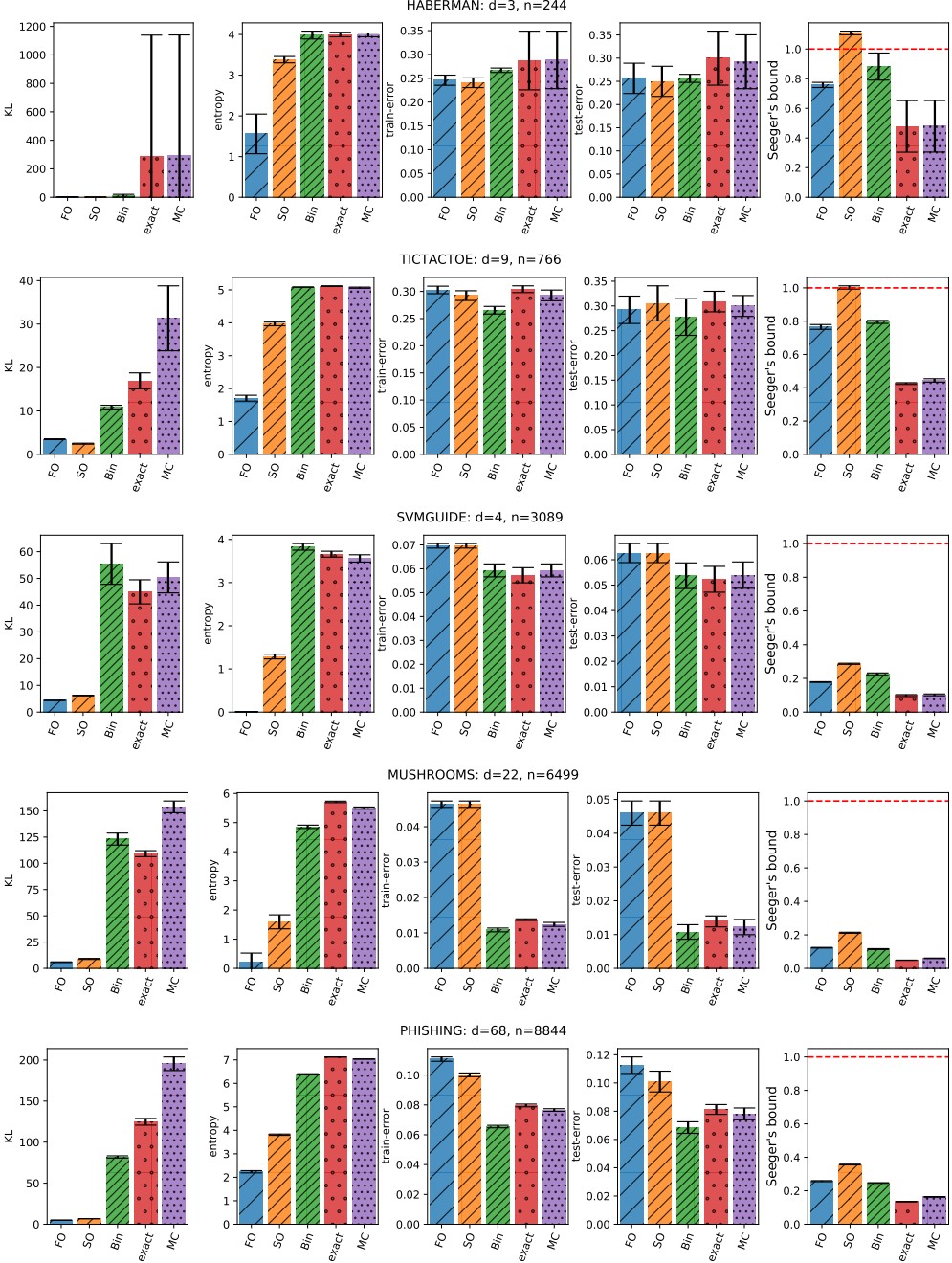

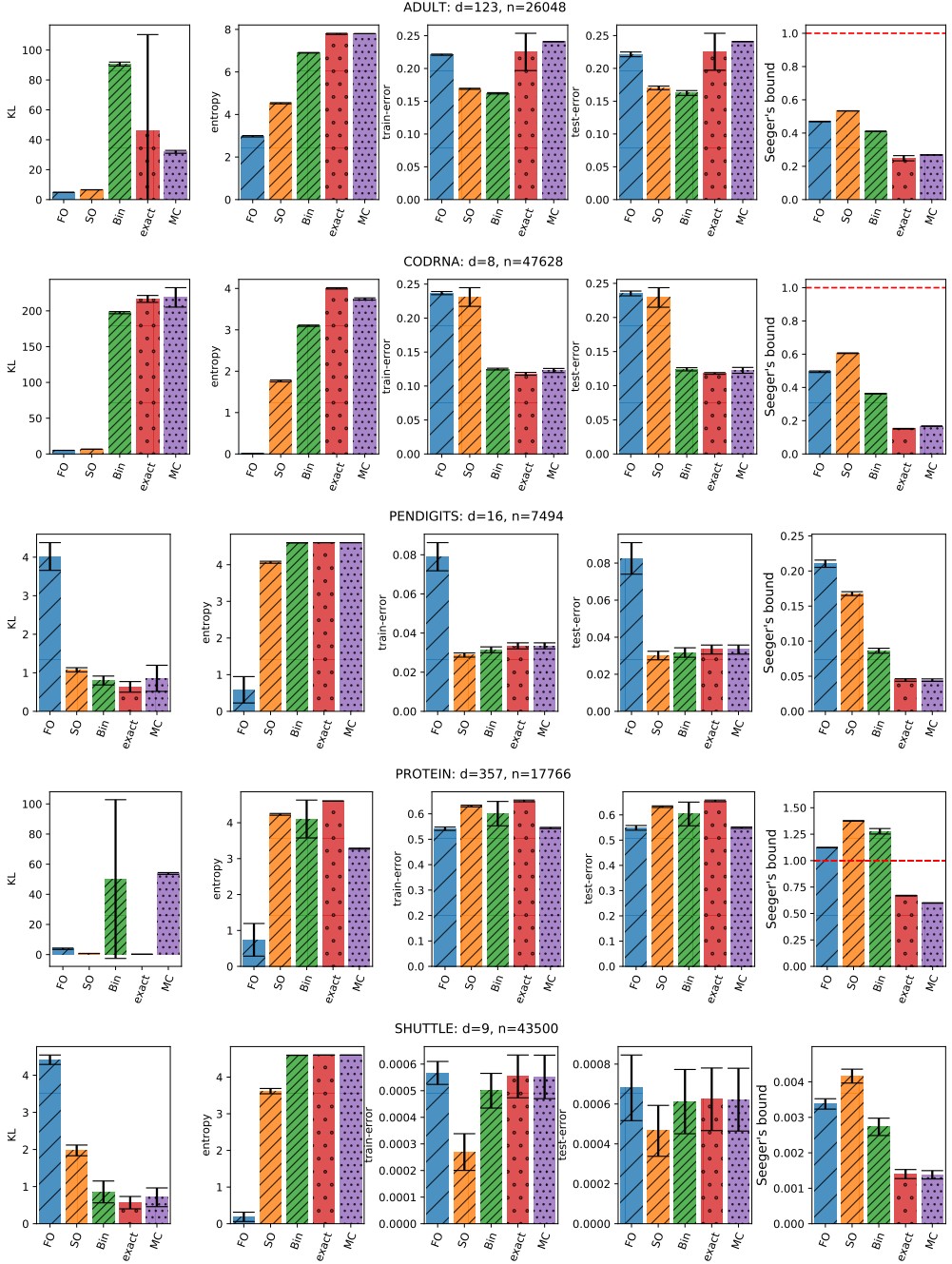

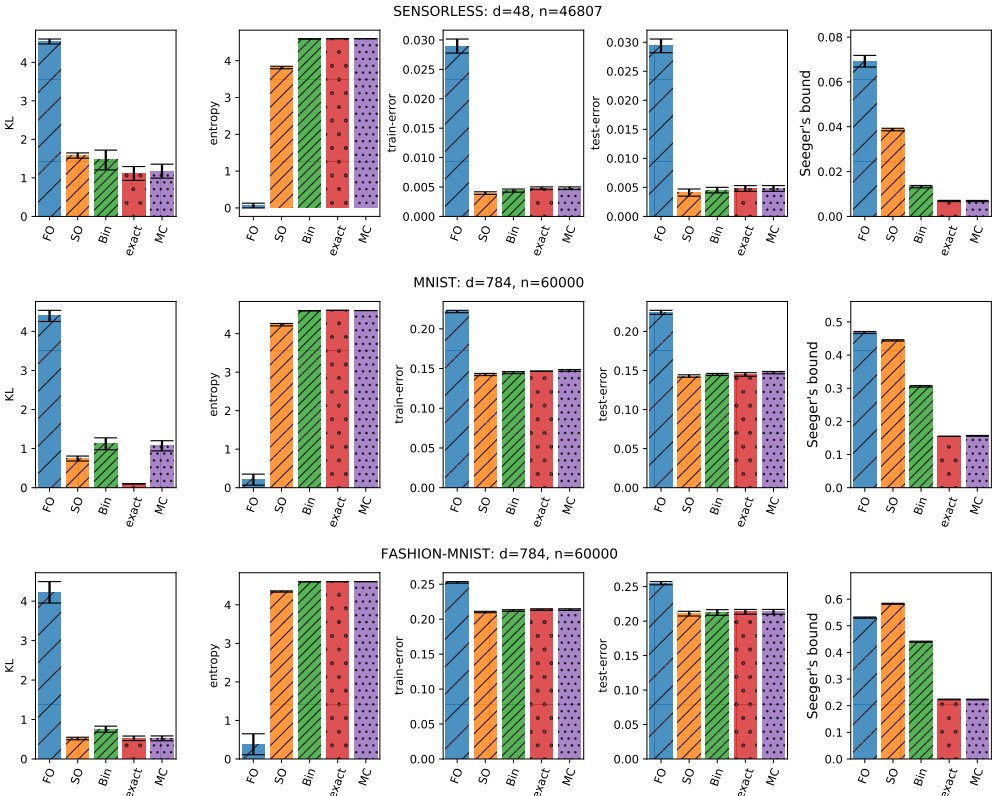

Figure 14: Comparison of First Order (*FO*), Second Order (*SO*) and Binomial (*Bin*) and our methods (*exact*, *MC*) in terms of training and test error rates and PAC-Bayesian bound values. For *exact* and *MC* the entropy is computed for the average MV given the learned Dirichlet distribution. Each row of subfigures corresponds to a dataset, where we marked its number of features $d$ and number of training instances $n$. The dashed horizontal line in the rightmost column plots marks the threshold above which the bounds are vacuous. We report the means (bars) and standard deviations (vertical, black lines) over 10 different runs.

Table 1: Summary of the main results on binary datasets with data agnostic prior. For the PAC-Bayesian methods, we report the performance of the model obtained after optimizing Seeger's Bound (Theorem (1)). Bayesian Naive Bayes (BayesianNB) corresponds to the weighting strategy proposed in Berend and Kontorovich [2015]. For each method we report the average ($\pm$ standard deviations) of the test errors and generalization bounds (for the PAC-Bayesian methods) over 10 runs. We bold the results (test error and bound value) that are significantly better (smaller) than the other results for a dataset. Notice that the PAC-Bayesian methods generally improve upon BayesianNB and that our method consistently provides the tightest and non-vacuous generalization bounds over all the datasets.

| | Method | HABERMAN | TICTACTOE | SVMGUIDE | MUSHROOMS | PHISHING | CODRNA | ADULT |
|---|---|---|---|---|---|---|---|---|
| Test error | Bayesian NB | 25.65 ± 2.10 | 30.21 ± 3.06 | 34.51 ± 0.00 | 11.65 ± 0.59 | 44.49 ± 0.05 | 66.67 ± 0.00 | 24.07 ± 0.00 |
| | First Order | 25.65 ± 3.26 | 29.22 ± 2.77 | 6.26 ± 0.37 | 4.60 ± 0.36 | 11.26 ± 0.58 | 23.52 ± 0.35 | 22.14 ± 0.36 |
| | Second Order | 25.00 ± 3.25 | 30.52 ± 3.55 | 6.26 ± 0.37 | 4.60 ± 0.36 | 10.10 ± 0.73 | 22.93 ± 1.43 | 17.02 ± 0.29 |
| | Binomial | 25.65 ± 0.87 | 27.76 ± 3.71 | **5.37 ± 0.50** | **1.08 ± 0.22** | **6.84 ± 0.41** | 12.41 ± 0.23 | **16.24 ± 0.37** |
| | ours-exact | 30.00 ± 5.83 | 30.88 ± 2.08 | **5.23 ± 0.51** | **1.39 ± 0.16** | 8.13 ± 0.35 | **11.80 ± 0.13** | 22.54 ± 2.81 |
| | ours-MC | 29.22 ± 5.81 | 30.00 ± 2.11 | **5.40 ± 0.52** | **1.22 ± 0.23** | 7.81 ± 0.42 | 12.25 ± 0.43 | 24.07 ± 0.00 |
| Seeger's b. | First Order | 75.76 ± 1.75 | 76.52 ± 1.44 | 17.84 ± 0.21 | 12.31 ± 0.19 | 25.75 ± 0.31 | 49.45 ± 0.47 | 46.93 ± 0.18 |
| | Second Order | 110.83 ± 1.24 | 100.25 ± 0.99 | 28.67 ± 0.33 | 21.27 ± 0.31 | 35.67 ± 0.21 | 60.59 ± 0.20 | 53.33 ± 0.09 |
| | Binomial | 88.17 ± 9.12 | 79.55 ± 0.90 | 22.52 ± 0.70 | 11.49 ± 0.27 | 24.64 ± 0.20 | 36.24 ± 0.13 | 41.06 ± 0.13 |
| | ours-exact | **47.83 ± 17.40** | **42.54 ± 0.52** | **9.79 ± 0.57** | **4.85 ± 0.09** | **13.49 ± 0.21** | **15.17 ± 0.21** | **24.87 ± 1.56** |
| | ours-MC | **47.89 ± 17.38** | 44.42 ± 1.02 | **10.21 ± 0.54** | 5.99 ± 0.16 | 16.34 ± 0.19 | 16.72 ± 0.12 | 26.83 ± 0.03 |

Table 2: Summary of the main results on multi-class datasets with informed priors. For the PAC-Bayesian methods, we report the performance of the model obtained after optimizing Seeger's Bound with informed priors (Theorem (2)). Bayesian Naïve Bayes (BayesianNB) corresponds to the weighting strategy proposed in Berend and Kontorovich [2015]. For each method we report the average ($\pm$ standard deviations) of the test errors and generalization bounds (for the PAC-Bayesian methods) over 10 runs. We bold the results (test error and bound value) that are significantly better (smaller) than the other results for a dataset. Notice that the PAC-Bayesian methods generally improve upon BayesianNB and that our method consistently provides the tightest and non-vacuous generalization bounds over all the datasets.

| | Method | PENDIGITS | PROTEIN | SHUTTLE | SENSORLESS | MNIST | FASHION-MNIST |
|---|---|---|---|---|---|---|---|
| Test error | Bayesian NB | 82.41 ± 0.11 | 82.09 ± 0.00 | 20.90 ± 0.03 | 86.27 ± 0.21 | 86.09 ± 0.09 | 89.14 ± 0.05 |
| | First Order | 8.25 ± 0.85 | **54.86 ± 0.93** | 0.07 ± 0.02 | 2.94 ± 0.12 | 22.44 ± 0.26 | 25.50 ± 0.23 |
| | Second Order | 3.01 ± 0.23 | 63.26 ± 0.30 | 0.05 ± 0.01 | 0.41 ± 0.06 | 14.29 ± 0.16 | 21.08 ± 0.34 |
| | Binomial | 3.17 ± 0.26 | 60.37 ± 4.69 | 0.06 ± 0.02 | 0.45 ± 0.05 | 14.49 ± 0.13 | 21.25 ± 0.41 |
| | ours-exact | 3.33 ± 0.24 | 65.54 ± 0.33 | 0.06 ± 0.02 | 0.48 ± 0.05 | 14.49 ± 0.23 | 21.34 ± 0.35 |
| | ours-MC | 3.33 ± 0.24 | **54.92 ± 0.28** | 0.06 ± 0.02 | 0.48 ± 0.05 | 14.73 ± 0.15 | 21.34 ± 0.35 |
| Seeger's b. | First Order | 21.04 ± 0.52 | 112.34 ± 0.25 | 0.34 ± 0.01 | 6.92 ± 0.26 | 46.81 ± 0.30 | 53.08 ± 0.21 |
| | Second Order | 16.77 ± 0.28 | 137.60 ± 0.26 | 0.42 ± 0.02 | 3.87 ± 0.06 | 44.44 ± 0.18 | 58.26 ± 0.19 |
| | Binomial | 8.64 ± 0.35 | 127.74 ± 2.59 | 0.27 ± 0.02 | 1.33 ± 0.05 | 30.59 ± 0.23 | 43.99 ± 0.23 |
| | ours-exact | **4.46 ± 0.18** | 66.86 ± 0.34 | **0.14 ± 0.01** | **0.69 ± 0.03** | **15.50 ± 0.02** | **22.35 ± 0.12** |
| | ours-MC | **4.45 ± 0.17** | **59.93 ± 0.24** | **0.14 ± 0.01** | **0.69 ± 0.03** | 15.64 ± 0.11 | **22.33 ± 0.12** |