# OpenReview forum: "Learning Stochastic Majority Votes by Minimizing a PAC-Bayes Generalization Bound"
_NeurIPS.cc/2021/Conference — NeurIPS 2021 Poster_

### Official Review · Reviewer_62DW · 2021-07-12

**Rating:** 7
**Confidence:** 4

**Summary:**

This paper studies the generalization properties of the stochastic weighted majority vote (MV) classifiers, where the weight of the MV follows another distribution over MVs. The proposal allows a direct analysis of the risk of the stochastic MV instead of analyzing through other empirical quantities, which leads to tighter bounds. By taking the distribution over MVs to be Dirichlet distributions, the authors also provide an exact and an approximation method to optimize the parameters of the distribution.

**Ethical Concerns:**

I don't see any so far.

**Limitations And Societal Impact:**

Yes. The authors have discussed the potentially large KL increasing with M. The runtime of the exact method could be large, while the runtime of the approximation method is small for smaller M but increases with M. On the other hand, I don't see any negative societal impact in this work so far.

**Main Review:**

Most of the previous results for weighted majority vote (MV) considered the performance of a deterministic MV. To the best of my knowledge, this is the first investigation of stochastic MVs. The idea might open a new path to investigate the performance of the ensembled classifiers. However, some refinement in the paper might be required.

Major points:

1) Equation (7) in the proof is incorrect, although the conclusion holds. The probability follows the distribution, but the equality doesn't hold. Also, Lemma 2 is important for the idea to work, but the reference or the proof of Lemma 2 is missing.

2) The reported generalization bounds for the new methods are perhaps not very comparable to the others in the sense that it is the generalization bound for the stochastic MV, which is different from the deterministic MVs. If we consider the deterministic MV induced by the stochastic MV, for example, the expected MV, the bound would be 2 times larger.


Minor points:

1) The notation of the vectors is not consistent throughout the paper. It would be better to use (), instead of [] or {} to denote a vector. "unit vector" in line 161 should be "all-ones vector." In the statement of Lemma 2, [1, M] should be [M]. The definition of R(\rho) in line 175 is missing. \hat{R}_{\leq m} (\hat{R}_{> m}), and their definitions in Line 233 are missing. \mathcal{D} in line 224 should be D. Also, some references in the appendix are not properly handled, for instance, the captions of Figure 9 and Table 1.

**Time Spent Reviewing:**

10

---

> ### Author Response · Authors · 2021-08-10
> **Corrections and clarification on comparison with deterministic baselines**
>
> "_Most of the previous results for weighted majority vote (MV) considered the performance of a deterministic MV. To the best of my knowledge, this is the first investigation of stochastic MVs. The idea might open a new path to investigate the performance of the ensembled classifiers. However, some refinement in the paper might be required._"
>
> We thank you for this very positive comment!
>
> "_Major points:
> Equation (7) in the proof is incorrect, although the conclusion holds. The probability follows the distribution, but the equality doesn't hold. Also, Lemma 2 is important for the idea to work, but the reference or the proof of Lemma 2 is missing._"
>
> Thank you for pointing these out. Indeed there is an inaccuracy in the notation for Equation (7) where the probability density follows the distribution but is of course not equal to it. In the revised version of the paper, we have fixed this notation and also referenced the proof for Lemma 2 (the aggregation property of Dirichlet distributions) available in Section 2.3.1 of [1].
>
> "_The reported generalization bounds for the new methods are perhaps not very comparable to the others in the sense that it is the generalization bound for the stochastic MV, which is different from the deterministic MVs. If we consider the deterministic MV induced by the stochastic MV, for example, the expected MV, the bound would be 2 times larger._"
>
> Because stochastic MV is a novel algorithm, the only baselines we can compare to are deterministic. We chose to report the performance of the models used in practice, which indeed means that for First Order, Second Order and RandomizedMV the bounds and test errors are for a deterministic MV, while for our method the bounds and test errors are for a stochastic MV. We stress that the scope of the paper is to make the optimization of PAC-Bayesian bounds tractable without resorting to upper bounds of the 01-loss. As a by-product, this work can also yield improved bounds for deterministic MVs, however this is not a central contribution. If we consider the average MV, the bounds are indeed 2 times larger than the ones reported in Figure 5 and 6 for exact and MC when using the inequality of line 240. These derandomized bounds are still tighter or comparable with First Order, Second Order and RandomizedMV’s ones, and could be further improved by using more advanced derandomization techniques, such as [2,3], which do not require a factor 2.
>
> "_Minor points:
> The notation of the vectors is not consistent throughout the paper. It would be better to use (), instead of [] or {} to denote a vector. "unit vector" in line 161 should be "all-ones vector." In the statement of Lemma 2, [1, M] should be [M]. The definition of R(\rho) in line 175 is missing. \hat{R}{\leq m} (\hat{R}{> m}), and their definitions in Line 233 are missing. \mathcal{D} in line 224 should be D. Also, some references in the appendix are not properly handled, for instance, the captions of Figure 9 and Table 1._"
>
> Thank you for pointing out these problems -- which are now corrected in the revised version of our manuscript.
>
> [1] Bela A. Frigyik, Amol Kapila, and Maya R. Gupta. "Introduction to the Dirichlet distribution and related processes." UWEE Technical Report Number UWEETR-2010-0006 (2010) https://vannevar.ece.uw.edu/techsite/papers/documents/UWEETR-2010-0006.pdf
>
> [2] Olivier Catoni. PAC-Bayesian Supervised Classification: The Thermodynamics of Statistical Learning. Institute of Mathematical Statistics Lecture Notes, 2007
>
> [3] Omar Rivasplata, Ilja Kuzborskij, Csaba Szepesvári, and John Shawe-Taylor. PAC-Bayes analysis beyond the usual bounds. In NeurIPS, 2020.

---

> > ### Comment · Reviewer_62DW · 2021-08-23
> > **Reviewer Response**
> >
> > Thank you for the response. I have decided to raise my score.

---

### Official Review · Reviewer_4Cot · 2021-07-14

**Rating:** 7
**Confidence:** 4

**Summary:**

The paper proposes a novel majority-vote learning algorithm for finite hypothesis classes based on minimizing a PAC-Bayes bound. The paper also reports experimental results that the theoretical generalization bound is close to the observed held-out error i.e., the generalization bound is tight, when the base hypothesis class is a collection of decision trees.

**Limitations And Societal Impact:**

There is at least one other work also involving PAC-Bayes majority vote and decision trees that might be relevant to your work https://papers.nips.cc/paper/2018/hash/1819020b02e926785cf3be594d957696-Abstract.html. This paper aims to derive classifiers whose excess risk (risk of classifier minus Bayes risk) goes to zero if the hypothesis class is allowed to grow with the dataset size (like decision trees of increasing depth). The tightness here is a different sense than the tightness in your paper, however.

On another note, are extensions to continuous hypotheses spaces possible? For instance, neural networks. These are the hypothesis classes where most of the interesting work in statistical learning is being done currently.

The societal impact has been addressed well.


**Main Review:**

Originality. Existing PAC-Bayes learning algorithms are designed to a) find a probability distribution Q over base hypotheses H and b) define the final majority-vote classifier as the one outputting the average prediction under Q. For finite hypothesis classes H, the paper’s learning algorithm modifies a) by implicitly redefining the base hypothesis class to be H’, where each element h’ is a convex combination of all the elements of H (Equation 4). With this H’, the paper returns to the standard methodology, by searching for Q’ by minimizing a generalization bound (either Theorem 1 or Theorem 2), and derandomizing to return the final majority-vote classifier. Section 3.1 and Section 3.2 explain that the generalization bound as a function of the parameters defining the distribution Q’ is differentiable. Therefore, first-order methods such as gradient descent can be used to reach (at least) local minima. More importantly, since the bounds hold for arbitrary Q’, it is always possible to evaluate performance even if training is not perfect (reaching global optima).

Quality. The claims in the paper are well-supported by evidence. I have some questions if time permits/for future work. First, can we inspect how different the final majority vote classifier in Figure 5 for exact is from the baselines? I am puzzled by how the different methods have similar test errors, but the generalization bounds are so different. Since the majority votes are defined by the convex combination with respect to the base hypotheses, this should be simple visualization. Second, is there a concrete, synthetic situation that gives intuition why the paper’s approach of first implicitly defining H’ and then computing Q’ gives better bounds then the standard PAC-Bayes approach? As an analog, section 2.2 lists some ideas of why a second order bound is better than a first-order one (because of correlation among the base hypotheses). Is there something like that for StocMC? Third, what do the different components of the generalization bounds before the de-randomization stage look like? This is related to my first question. All the PAC-Bayes bounds balance a trade off between empirical fit (the Rhat term) with complexity (the KL term). Since the different methods in Figure 5 have similar test error, I am tempted to think that the difference in generalization bounds come from the complexity term. In any case it would be good to check!

Clarity. The paper is clear and well-written. One suggestion is putting the datasets where both the test error and the generalization bounds are too small (like sensorless, mushrooms) to the appendix.

Significance. The consistent ability of the generalization bound to approximate the test error from above (Figure 5) is very encouraging evidence that this kind of majority-vote algorithm can explain how ensemble methods generalize on unseen data.

**Time Spent Reviewing:**

3

---

> ### Author Response · Authors · 2021-08-10
> **Comments on suggested improvements**
>
> "_Quality. The claims in the paper are well-supported by evidence. I have some questions if time permits/for future work. First, can we inspect how different the final majority vote classifier in Figure 5 for exact is from the baselines? I am puzzled by how the different methods have similar test errors, but the generalization bounds are so different. Since the majority votes are defined by the convex combination with respect to the base hypotheses, this should be simple visualization. Second, is there a concrete, synthetic situation that gives intuition why the paper’s approach of first implicitly defining H’ and then computing Q’ gives better bounds then the standard PAC-Bayes approach? As an analog, section 2.2 lists some ideas of why a second order bound is better than a first-order one (because of correlation among the base hypotheses). Is there something like that for StocMC? Third, what do the different components of the generalization bounds before the de-randomization stage look like? This is related to my first question. All the PAC-Bayes bounds balance a trade off between empirical fit (the Rhat term) with complexity (the KL term). Since the different methods in Figure 5 have similar test error, I am tempted to think that the difference in generalization bounds come from the complexity term. In any case it would be good to check!_"
>
> Thank you for your overall positive assessment, and for all these interesting suggestions. Empirically, we checked the different components of the bounds (e.g. train loss, complexity) and indeed we agree it would be valuable to report all these values in the paper: should our paper be accepted we will gladly incorporate this. What we saw is that the principal source of looseness of state-of-the-art bounds for MVs are the irreducible factors stemming from the use of an upper bound. Figure 1 provides a graphical proof for this. State-of-the-art bounds rely on surrogate losses (upper bounds of the error rate) while our method optimizes directly the 01-loss. In other words, this means that all the baselines optimize a subset of the statistical moments of the error (First Order only the first moment; C-Bound both the first and the second so it can account for correlations). On the contrary, our method optimizes the exact problem, which implies that for any set of base classifiers (more or less diverse/correlated) it will always estimate the tightest guarantee.
>
> "_Clarity. The paper is clear and well-written. One suggestion is putting the datasets where both the test error and the generalization bounds are too small (like sensorless, mushrooms) to the appendix.
> Significance. The consistent ability of the generalization bound to approximate the test error from above (Figure 5) is very encouraging evidence that this kind of majority-vote algorithm can explain how ensemble methods generalize on unseen data._"
>
> We thank you for this very positive comment!
>
> "_Limitations And Societal Impact:
> There is at least one other work also involving PAC-Bayes majority vote and decision trees that might be relevant to your work https://papers.nips.cc/paper/2018/hash/1819020b02e926785cf3be594d957696-Abstract.html. This paper aims to derive classifiers whose excess risk (risk of classifier minus Bayes risk) goes to zero if the hypothesis class is allowed to grow with the dataset size (like decision trees of increasing depth). The tightness here is a different sense than the tightness in your paper, however.
> On another note, are extensions to continuous hypotheses spaces possible? For instance, neural networks. These are the hypothesis classes where most of the interesting work in statistical learning is being done currently.
> The societal impact has been addressed well._"
>
> Thank you for the suggestion. We will gladly discuss this paper (of which we were not aware, many thanks for pointing this out!). As you mention, [Nguyen and Kpotufe] paper does not address the same settings nor use the same tools, but we agree it is relevant as it provides PAC-Bayes (excess risk) bounds for a particular decision tree based ensemble strategy. We will carefully highlight the differences of approaches to the reader. Regarding your last point, we believe that our contributions are mainly for finite hypothesis sets. However, this set is not restricted to classical models and can also be defined as an ensemble of deep neural networks models, as an example, which are used in practical applications to achieve high accuracy, robustness and for uncertainty estimation. Extending this work to continuous hypothesis spaces is indeed an exciting venue for future work.

---

### Official Review · Reviewer_t4tj · 2021-07-16

**Rating:** 5
**Confidence:** 3

**Summary:**

The paper gives a tighter bound on the generalization of majority voting classifier by using a form of stochastic majority vote classifier. Using minimization of this bound, they are able to get better accuracy for the classifier as well as tighter generalization guarantees.


**Limitations And Societal Impact:**

Yes

**Main Review:**

The paper uses PAC Bayesian bounds on stochastic majority voting scheme to get tighter generalization bounds on its performance. The paper is interesting and they are able to achieve good results experimentally. But, my main concern is that the main contribution of the paper is to get tighter generalization bounds by using stochastic majority votes. However, it is not clear to me why the bound is tighter when going from the error of one classifier to the expected majority vote classifier. On line 240, this conversion still introduces a factor of 2. So, it is not clear why this bound is tighter. The statement made in the paragraph is vague and hard to understand.

Can the authors clarify the statement made in line 141 that second order and C-bounds fail to leverage the diversity in the whole set of classifiers?

The paper seems interesting to me. However, I feel that the paper is not well written and many claims are vague without proper explanation and thus, hard to understand. The writing of the paper can be improved.

**Time Spent Reviewing:**

3

---

> ### Author Response · Authors · 2021-08-10
> **Clarification on derandomized bound and base classifier diversity**
>
> "_The paper uses PAC Bayesian bounds on stochastic majority voting schemes to get tighter generalization bounds on its performance. The paper is interesting and they are able to achieve good results experimentally. But, my main concern is that the main contribution of the paper is to get tighter generalization bounds by using stochastic majority votes. However, it is not clear to me why the bound is tighter when going from the error of one classifier to the expected majority vote classifier. On line 240, this conversion still introduces a factor of 2. So, it is not clear why this bound is tighter. The statement made in the paragraph is vague and hard to understand._"
>
> To clarify, in line 240 we reported an example of possible derandomization: it introduces a global factor 2 but no additional factors on the KL term, such as Second Order which has a 2 KL() term and RandomizedMV which has a N KL() term. Tighter derandomized PAC-Bayesian guarantees (resorting to the results of [1, 2]) can also be derived, which bound the risk of a sampled MV and do not require a factor 2. We stress that the scope of the paper is to make the optimization of PAC-Bayesian bounds tractable without resorting to upper bounds of the 01-loss. As a by-product, this work can also yield improved bounds for deterministic MVs, however this is not a central contribution.
>
> "_Can the authors clarify the statement made in line 141 that second order and C-bounds fail to leverage the diversity in the whole set of classifiers?_"
>
> By diversity we mean that base classifiers assign to one class or the other different sets of points, hence make mistakes on different points (i.e. they disagree). In Figure 3, even though the base classifiers are weak, they are diverse enough so that there exists an optimal combination of them that  perfectly splits the two classes without error. However, the optimization of the PAC-Bayes guarantees over First Order, Second Order and C-Bound are shown to select a small subset of base classifiers which is not enough to achieve good performance.
>
> "_The paper seems interesting to me. However, I feel that the paper is not well written and many claims are vague without proper explanation and thus, hard to understand. The writing of the paper can be improved._"
>
> We hope that we managed to address the specific points raised in the review, by clarifying the confusing statements. If the reviewer would like to specifically point out other unclear parts of the paper, we would be happy to be given the chance to further clarify them.
>
> [1] Olivier Catoni. PAC-Bayesian Supervised Classification: The Thermodynamics of Statistical Learning. Institute of Mathematical Statistics Lecture Notes, 2007
>
> [2] Omar Rivasplata, Ilja Kuzborskij, Csaba Szepesvári, and John Shawe-Taylor. PAC-Bayes analysis beyond the usual bounds. In NeurIPS, 2020.

---

### Official Review · Reviewer_pW3i · 2021-07-18

**Rating:** 7
**Confidence:** 4

**Summary:**

The authors propose an algorithm for learning a *stochastic* weighted majority vote classifier, by optimizing a (PAC-Bayesian) generalization bound. They consider Dirichlet distributions over classifiers as well as general measures, and derive analytical and numerical schemes, respectively, for optimizing both. They empirically evaluate the algorithm for several datasets and model classes (decision stumps and random forest). They also compare against deterministic weighted majority vote classifiers learned by optimizing Gibbs classifiers via existing PAC-Bayes bounds (first order, secord order, C-bounds, etc). The resulting stochastic predictor achieves competitive performance and bound tightness compared to ones obtained by minimizing alternative bounds.

**Ethical Concerns:**

This work does not raise any ethical concerns.

**Limitations And Societal Impact:**

The authors discuss the following limitations of the proposed algorithm: the algorithm seems to underperform under input noise; the obtained guarantees get looser with an increasing number of base predictors. These seem to be fair limitations.

The authors do not foresee any negative societal impact.

**Main Review:**

In my opinion, the paper presents fairly incremental ideas, combining data-dependent priors, a few other existing tricks in PAC-Bayes, like splitting the datasets in half, etc. From the technical perspective, the authors work out a closed form for stochastic Dirichlet predictors. This is a nice contribution, but appears to be relatively straightforward. Since the non-Dirichlet approach is competitive, it is not clear to me how useful the derived analytical expressions are. As the authors report, the MC variant seems to be fast and accurate with a range of base predictors.

However, the empirical results are positive and the empirical evaluation is, I believe, novel. The clarity and quality are both good. Overall, I am leaning towards acceptance but would like to see authors’ responses to a few questions listed below (especially about the advantage of stochastic majority vote classifiers over non-stochastic counterparts).

What are SOTA algorithms for optimizing standard (deterministic) weighted majority vote classifiers? The comparisons here are all to PAC-Bayes bound optimization, but are these algorithms at all competitive? And how about the same question but with stochastic majority vote classifiers? I would like the authors to provide references and report their (test) performance in the same cases as tested in Figure 5. How do they compare in error? If these other algorithms do better, what is the advantage of adding stochasticity? How do PAC-Bayes bounds on error of stochastic majority vote compare to held-out set bounds for non-stochastic majority vote achieved by these alternative algorithms?

To what extent does the empirical evaluation replicate what has already been reported in previous papers (e.g., on first and second order bounds)? That is: is the empirical evaluation performed here to be considered a novel contribution?

When trained on the datasets used in Fig 5 (so not just the moon dataset), did you also observe that Dirichlet had high entropy/higher than for classifiers trained with alternative algorithms?

The authors also present a hypothesis why their algorithm’s performance (or tightness of the bound?) decreases (relative to other algorithms tested) under input noise (using 0-1 vs loss with margin information). While a theoretical analysis might be challenging, it would be nice to see some empirical evidence evaluating their hypothesis.

Also, what is the (approximate, perhaps) Bayes error for the noise levels used in Figure 12? What should we compare the performance against?

**Time Spent Reviewing:**

8

---

> ### Author Response · Authors · 2021-08-10
> **Clarification of significance and novelty of the approach and results**
>
> "_In my opinion, the paper presents fairly incremental ideas, combining data-dependent priors, a few other existing tricks in PAC-Bayes, like splitting the datasets in half, etc. From the technical perspective, the authors work out a closed form for stochastic Dirichlet predictors. This is a nice contribution, but appears to be relatively straightforward. Since the non-Dirichlet approach is competitive, it is not clear to me how useful the derived analytical expressions are. As the authors report, the MC variant seems to be fast and accurate with a range of base predictors._"
>
> We understand that our work might appear incremental as the derivation of the bounds are based on classical or recent works in PAC-Bayes. However, setting up the problem was not straightforward and the results are certainly not trivial. The proposed model, the stochastic MV, is an original contribution (as also pointed out by reviewer 62DW) allowing us to derive tight PAC-Bayesian guarantees without resorting to upper bounds of the 01-loss. Moreover, the derived closed-form allows us to optimize the 01-loss (or error rate) exactly for the particular choice of Dirichlet distributions, which is intractable in most cases. Regarding the practical usefulness of this closed-form, indeed the MC variant is able to obtain close to optimal solutions in most of the studied settings. However, the MC variant is based on a relaxation of the problem, hence it is not guaranteed to attain the optimal solution in other contexts not considered in our experimental analysis. As a general rule of thumb, we would rather leave it to the practitioner to choose the right balance between computational cost and accuracy, rather than imposing this choice: the optimality of the relaxation depends on the problem at-hand and we do not claim that this relaxation is universally good beyond the settings we explored. We hope that this brings clarity as to when our strategies are most useful.
>
> "_However, the empirical results are positive and the empirical evaluation is, I believe, novel. The clarity and quality are both good. Overall, I am leaning towards acceptance but would like to see authors’ responses to a few questions listed below (especially about the advantage of stochastic majority vote classifiers over non-stochastic counterparts).
> What are SOTA algorithms for optimizing standard (deterministic) weighted majority vote classifiers? The comparisons here are all to PAC-Bayes bound optimization, but are these algorithms at all competitive? And how about the same question but with stochastic majority vote classifiers? I would like the authors to provide references and report their (test) performance in the same cases as tested in Figure 5. How do they compare in error? If these other algorithms do better, what is the advantage of adding stochasticity? How do PAC-Bayes bounds on error of stochastic majority vote compare to held-out set bounds for non-stochastic majority vote achieved by these alternative algorithms?_"
>
> Figure 3, and Tables 1 and 2 in the appendix report the results also of a non-PAC-Bayesian strategy for learning MV:  Bayesian Naive Bayes proposed in [1]. Previous works, such as [3](Table 1) and [4] (Figure 4), have extensively compared PAC-Bayesian techniques to boosting methods and shown that they achieve comparable accuracy, apart from providing generalization guarantees.
> As the stochastic counterpart of MV is a novel contribution, to the best of our knowledge, we cannot report a comparison with other stochastic MV techniques. The closest related works are the reported First Order, Second Order and Randomized MV but they all are deterministic models. The main advantage of adding stochasticity is to enable the optimization of the 01-loss and the derivation of tighter PAC-Bayesian bounds. The reason why the bounds are tighter is graphically shown in Figure 1, where we see that the objective functions of all the baselines are upper bounds of the 01-loss.
>
> "_To what extent does the empirical evaluation replicate what has already been reported in previous papers (e.g., on first and second order bounds)? That is: is the empirical evaluation performed here to be considered a novel contribution?_"
>
> While First Order is a classical technique and has been extensively studied, also under the settings in our experimentation, Second Order has been recently published in [2] and studied only in the context of strong base classifiers (i.e. data-dependent decision forests), to the best of our knowledge. The comparison of First Order and Second Order with weak base classifiers (i.e. decision stumps) or with varying maximal tree depth is new and sheds light into why FO surprisingly provides the best generalization guarantees for deterministic MVs while its test error is generally higher.
> Other new elements in our empirical evaluation are the study of the robustness of PAC-Bayesian MVs to input noise, and, of course, the additional comparison with the exact and MC variants of our method.
>
> "_When trained on the datasets used in Fig 5 (so not just the moon dataset), did you also observe that Dirichlet had high entropy/higher than for classifiers trained with alternative algorithms?
> The authors also present a hypothesis why their algorithm’s performance (or tightness of the bound?) decreases (relative to other algorithms tested) under input noise (using 0-1 vs loss with margin information). While a theoretical analysis might be challenging, it would be nice to see some empirical evidence evaluating their hypothesis.
> Also, what is the (approximate, perhaps) Bayes error for the noise levels used in Figure 12? What should we compare the performance against?_"
>
> Thank you for all these interesting suggestions! Following on those, we have checked on the datasets on Figure 5a the entropy of the deterministic models compared to the entropy of the average MV given the learned Dirichlet distribution. The entropy is consistently higher for our model. More precisely, by noting by H() the entropy, we see that generally H(First Order) < H(Second Order) < H(Randomized MV) < H(ours).
> Concerning the sensitivity of our method to input noise,  Figure 14 in the appendix partially provides empirical evidence to support our hypothesis that points in the considered settings have margins close to 0.5. In the first column, we report (as a function of the maximal tree depth) the voter strength, expressed as the average of the margin distribution for a dataset. Although the mean value does not provide all the information of a distribution, this study suggests that margin distributions probably have values around 0.5. We plan to extend the study of the margin distributions, e.g. by plotting their histograms and checking other statistical moments, to see if and how these affect the accuracy of our model. This study will indeed complement the preliminary analysis reported in Figure 12, whose aim is to see whether our method is sensitive to input noise and at which level of noise its error rate stops being lower than those of the baselines. In the revised version of the paper, we have reported an approximation of the Bayes error, which is higher than 0 for noise above 0.1, and almost 1 (the two classes almost completely overlap) for noise=1.
>
> [1] Berend, Daniel, and Aryeh Kontorovich. A finite sample analysis of the Naive Bayes classifier. In JMLR, 2015.
>
> [2] Andrés R. Masegosa, Stephan Sloth Lorenzen, Christian Igel, and Yevgeny Seldin. Second order PAC-Bayesian bounds for the weighted majority vote. In NeurIPS, 2020.
>
> [3] Roy, Jean-Francis, François Laviolette, and Mario Marchand. From PAC-Bayes bounds to quadratic programs for majority votes. In ICML. 2011.
>
> [4] Bauvin, Baptiste, et al. Fast greedy C-bound minimization with guarantees. Machine Learning 109.9. 2020.

---

> > ### Comment · Reviewer_pW3i · 2021-08-24
> > **Comments on new experiments and a question**
> >
> > Thank you for clarifying how the empirical evaluation of First and Second order bounds compares to previous work. It sounds like the comparison presented in this paper is novel and interesting.
> >
> > I appreciate the authors running extra experiments comparing the entropy of MV. The results seem to be consistent with the observations reported on the moon dataset.
> >
> > I personally think that optimizing relaxations or upper-bounds on 0-1 loss is not that problematic (since once can compute / report bounds on the desired 0-1 loss function), mostly because I have not encountered scenarios where there would be a huge discrepancy between the learned predictors from these different objective choices. I would personally be curious to hear from the authors whether they have run into practical datasets or can design a synthetic (but yet somewhat realistic) dataset where the closed-form expression for the opt objective makes a difference.
> >
> > Overall, I think the paper presents some interesting ideas and empirical observations are noteworthy and worth a publication.

---

> > > ### Author Response · Authors · 2021-08-27
> > > **thank you for the positive assessment!**
> > >
> > > The reviewer is right that good predictor accuracy is achievable by optimizing a relaxation of the zero-one loss. This is exactly what most of the usual learning algorithms perform on classification tasks. However, in the quest of designing a “self-bounding” learning algorithm that provides the tightest PAC-Bayesian generalization bounds together with accurate predictors, the classical choices of upper bounds on the majority vote risk may lead to a suboptimal behavior either in terms of the test loss or the bound value (or both).
> > >
> > > The extreme case occurs when applying the first order bound to weak predictors, as displayed by Figure 3(b): The linear loss is unable to capture the correlation between the voters. The obtained predictor would be as bad if we minimize solely the empirical linear loss, without the regularizing effect of the PAC-Bayes bound objective function. The same phenomenon is observable on many practical datasets when relying on weak voters like decision stumps.
> > >
> > > More clever choices of upper bounds on the majority vote lead to effective learning algorithms. That being said, as reported by Figures 4, 5 and 6, our “exact” method consistently provide tighter upper bounds than the Monte Carlo approximation and other compared methods.
> > >
> > > We hope that this answer the reviewer's question, and we would be happy to provide additional details if needed.

---

### Author Response · Authors · 2021-08-10
**General answer**

We warmly thank all reviewers for their careful evaluation of our paper. We feel encouraged by the overall positive tone of the four reviews which highlighted the significance of the problem, the novelty of our approach, the soundness of the theoretical and empirical analyses, and the clarity of the manuscript. We are particularly encouraged that, as noted by reviewers 4Cot and 62DW, our novel algorithm is seen as a promising path for the prominent problem of analyzing generalization of ensemble methods.  We are grateful for the diverse feedback and suggestions that greatly helped us to improve on our manuscript. We address the few concerns raised in the reviews and provide clarifications in specific answers.

---

### Decision · Program_Chairs · 2021-09-27

**Decision:**

Accept (Poster)

**Comment:**

The reviewers like the approach proposed in the paper and recommend acceptance. There were several clarity issues raised during the discussion and I hope they will be addressed in the final revision. In particular, the fixes to benchmark experiments that were discussed and revision of the two-moon experiment.